



# Relevance of aboveground litter for soil organic matter formation – a soil profile perspective

Patrick Liebmann[1], Patrick Wordell-Dietrich[2], Karsten Kalbitz[2], Robert Mikutta[3], Fabian Kalks[4], Axel Don[4], Susanne K. Woche[1], Leena R. Dsilva[2], Georg Guggenberger[1]

[1]Institute of Soil Science, Leibniz Universität Hannover, Herrenhäuser Str. 2, 30419, Hannover, Germany
[2]Institute of Soil Science and Site Ecology, Technische Universität Dresden, Pienner Straße 19, 01737 Tharandt, Germany
[3]Soil Science and Soil Protection, Martin Luther University Halle-Wittenberg, Von-Seckendorff-Platz 3, 06210 Halle (Saale), Germany
[4]Thünen Institute of Climate-Smart Agriculture, Bundesallee 65, 38116 Braunschweig, Germany

*Correspondence to*: Patrick Liebmann (liebmann@ifbk.uni-hannover.de)

**Abstract.** In contrast to mineral topsoils, the origin and processes leading to the formation and stabilization of organic matter (OM) in subsoils is still not well known. This study addresses the fate of litter-derived carbon (C) in whole soil profiles with regard to the conceptual cascade model, which proposes that OM formation in subsoils is linked to sorption-microbial processing-remobilization cycles during the downward migration of dissolved organic carbon (DOC). Our main objectives were to quantify the contribution of recent litter to subsoil C stocks via DOC movement and to evaluate the stability of litter-derived OM in different functional OM fractions.

A plot-scale stable isotope labeling experiment was conducted in a temperate beech forest by replacing the natural litter layer with $^{13}$C enriched litter on an area of 20 m$^2$ above a Dystric Cambisol. After 22 months of field exposure, the labeled litter was replaced again by natural litter and soil cores were drilled down to 180 cm soil depth. Water extraction and density fractionation were combined with stable isotope measurements in order to link the fluxes of recent litter-derived C to its allocation into different functional OM fractions. A second sampling was conducted 18 months later to further account for the stability of translocated young litter-derived C.

Almost no litter-derived particulate OM (POM) entered the subsoil, suggesting root biomass as the major source of subsoil POM. The contribution of aboveground litter to the formation of mineral-associated OM (MAOM) in topsoils (0-10 cm) was 0.99 ± 0.45 g C m$^{-2}$ yr$^{-1}$, and decreased to 0.37± 0.10 g C m$^{-2}$ yr$^{-1}$ in the upper subsoil (10-50 cm) and 0.01 ± 0.01 g C m$^{-2}$ yr$^{-1}$ in the deep subsoil > 100 cm soil depth. This finding suggests a subordinate importance of recent litter layer inputs via DOC translocation to subsoil C stocks, and implies that most of the OM in the subsoil is of older age. Smaller losses of litter-derived C within MAOM of about 66 % compared to POM (77–89 %) indicate that recent carbon can be stabilized by interaction with mineral surfaces; although the overall stabilization in the sandy study soils was low. Our isotope labeling approach supports the concept of OM undergoing a sequence of cycles of sorption, microbial processing, and desorption while migrating down a soil profile, which needs to be considered in models on soil OM formation and subsoil C cycling.





## 1 Introduction

The capability of soils to incorporate and preserve large quantities of organic matter (OM) is a key function in the global carbon (C) cycle (Wiesmeier et al., 2019). While in the past most studies on carbon inventories focused on topsoils, only some recent research also expands to subsoil environments (Fontaine et al., 2007; Salomé et al., 2010; Bernal et al., 2016), considering that a significant proportion of soil OM (SOM) is stored in subsoil horizons (Batjes, 1996; Jobbagy and Jackson, 2000). Major pathways of OM to enter subsoils are rhizodeposition, root exudation and dissolved organic matter (DOM) leached from the horizons above (Wilkinson et al., 2009; Rumpel and Kögel-Knabner, 2011; Kaiser and Kalbitz, 2012). Dissolved organic matter was estimated to contribute about 19 to 50 % to the total mineral soil C stock in forest soils (Kalbitz and Kaiser, 2008; Sanderman and Amundson, 2008) and is considered as main source of subsoil OM in temperate forest soils (Kaiser and Guggenberger, 2000). Further, its high affinity towards reactive mineral phases, thus forming mineral-associated OM (MAOM) makes DOM an important contributor to stabilized SOM (Leinemann et al., 2016).

Kaiser and Kalbitz (2012) described the interaction of OM with minerals as a sequence of processes including DOM sorption, microbial processing, and desorption, often referred to as the "cascade model". This model not only accounts for changes in dissolved organic carbon (DOC) concentration and bioavailability with depth, but also considers the depth-dependent changes in $^{14}$C age of SOM (Trumbore et al., 1992) as well as in DOM and SOM composition from plant- towards microbial-derived OM, likewise found in e.g. forest soils (Guggenberger and Zech, 1994; Kaiser et al., 2004). The cascade model also points at a microbial impact on exchange reactions of OM at mineral surfaces, which has been recently confirmed in a laboratory percolation experiment (Leinemann et al., 2018). Modern $^{14}$C ages of MAOM in mineral topsoil horizons, where most sorption sites are likely already occupied, also suggest such exchange of OM (Angst et al., 2018). Increasing OM degradation and transformation with soil depth often result in changes in the stable isotopic composition of SOM. In most soils, $\delta^{13}$C values increase with soil depth, which is related to the isotopic discrimination of the heavier C isotopes during microbial respiration (Nadelhoffer and Fry, 1988; Balesdent et al., 1993; van Dam et al., 1997). This depth trend can also reflect a translocation of relatively $\delta^{13}$C-enriched OM to greater depth due to preferential sorption of the $\delta^{13}$C-depleted carboxylated lignin degradation products via multiple sorption-decomposition-desorption steps (Kaiser et al., 2001). On the other hand, Rumpel et al. (2012) questioned the slow turnover of subsoil OM, since DOC and root exudate fluxes can substantially increase the subsoil C pool within decades—a view which is in contrast to the frequently high $^{14}$C ages of subsoil OM.

While the qualitative aspects of subsoil C cycling with respect to possible OM sources and processes are known (e.g. summarized by Schmidt et al. (2011) and Rumpel et al. (2012)), this does not refer to the controlling mechanisms and the turnover of the different subsoil C fractions. Assessment of OM turnover in the subsoil under real conditions still remains a major challenge, as it has to involve analysis of the different C sources (plant- versus microbial-derived) and the quantification of respective in- and outputs. In order to quantify individual C fractions and fluxes, stable isotope labeling, e.g. using $^{13}$C-enriched litter material, has been proven as a very powerful tool (Bird et al., 2008; Moore-Kucera and Dick, 2008). Yet, to the



best of our knowledge, there are no field studies available that employed stable isotope tracing to estimate the contribution of recent aboveground litter to subsoil C cycling. Also the role of recent litter-derived DOM in the formation of MAOM in the soil profile has not been quantified so far, nor has been the stability of the newly formed C fraction against microbial decomposition determined.

This study therefore addresses the fate of litter-derived C in the subsoil with regard to the conceptual cascade model
(Kaiser and Kalbitz, 2012) under field conditions. Our main objectives were to quantify the contribution of recent litter to subsoil C stocks via DOM movement and to evaluate the stability of litter-derived OM in different functional OM fractions. We aim at testing the following hypotheses:

1.    Recent aboveground litter does not contribute much to the subsoil C stocks due to retention and decomposition processes in the topsoil.

2.    Mineral-associated OM in the subsoil is primarily microbial-derived as a result of sorption-microbial processing-desorption cycles.

3.    Preservation of recent aboveground litter-derived C will be more pronounced when associated with minerals as compared to particulate organic matter (POM) fractions.

To test the hypotheses, we carried out a $^{13}$C-labeling experiment, where the natural litter layer on an acidic Cambisol
underneath European beech was replaced by a $^{13}$C-enriched leaf litter. The contribution of litter to subsoil OM was assessed by $\delta^{13}$C analysis in soil cores down to 180 cm soil depth sampled 22 and 40 months after field labeling. The labeled litter was changed back to unlabeled litter before sampling of the first cores allowing an indication of exchange processes of labeled C in the soil in the subsequent 18 months. Soil density fractionation was used to assess the contribution of young DOM to the formation of MAOM and to differentiate between particulate and dissolved pathway in the contribution of litter-derived OM
to subsoil OM.

## 2 Materials and methods

### 2.1 Site description

The field experiment was carried out in the Grinderwald beech forest (*Fagus sylvatica*), 40 km north of Hanover, Germany (52°34'22'' N, 9°18'49'' E) comprising a stand age of ca. 103 years. The common soil type in the research area is a Dystric
Cambisol (IUSS Working Group WRB, 2014), which developed from periglacial fluviatile sandy deposits. The mean annual temperature is 9.7°C and the mean annual precipitation is 762 mm (Deutscher Wetterdienst, Nienburg, 1981-2010). Selected soil properties of the Grinderwald sites are given in Table 1. More detailed site descriptions can be found in Angst et al. (2016) and Bachmann et al. (2016).





## 2.2 Experimental set-up

The study site Grinderwald includes three soil observatories on which $^{13}C$-labeled beech litter was applied (Leinemann et al., 2016; Wordell-Dietrich et al., 2019); hereafter referred to as plots 1 to 3. Each plot was divided in two compartments of 6.57 $m^2$ each. One compartment was labeled with $^{13}C$-enriched litter and the other one remained unlabeled as control. The experiment started in January 2015. For the labeling, the natural litter layer was removed manually and replaced by an equivalent amount of $^{13}C$ enriched beech litter, prepared as a mixture of highly labeled beech litter (10 atom-%, IsoLife,

Wageningen, The Netherlands) and unlabeled beech litter, which resulted in a final $^{13}C$ enrichment of 1241 to 1880 ‰ (Wordell-Dietrich et al., 2019). To avoid dilution of the labeled litter over time, a mesh (20 mm mesh size) was installed on top of the litter layer to remove seasonally fallen litter. The labeled litter stayed in the field for 22 months. In November 2016, the labeled litter was removed manually from all observatories.

Following the removal of the labeled litter, three soil cores per plot and treatment (labeled versus unlabeled) were taken

down to a depth of 200 cm using a machine-driven percussion coring system (Nordmeyer Geotool, Berlin, Germany). Since it was not possible for each soil core to secure the lowest increment of 180-200 cm, this depth was rejected from further processing. The cores were divided into 15 increments, starting with 5 cm increments from 0 to 10 cm, 10 cm increments from 10 to 100 cm, and 20 cm increments from 100 to 180 cm. Depth increments of the soil cores taken from 0-5 and 5-10 cm are defined as "topsoil", increments between 10 and 50 cm as "upper subsoil", those between 50 to 100 cm as "mid subsoil", and

increments below 100 cm as "deep subsoil". Directly after sampling, an equivalent amount of the natural beech litter of the surrounding area was used for replacement of the litter that has been removed before. A second sampling was conducted 18 months later, in May 2018, in total 40 months after applying the labeled litter on the plots.

Soil samples were oven-dried at 60°C and sieved < 2 mm. Three replicates per plot and treatment were combined to one composite sample per depth increment on a mass equivalent basis for further processing. Aliquots for water extractions

were stored frozen (-20°C) directly after sampling.

## 2.3 Analysis of bulk soil

Bulk samples were analyzed for organic C (OC), total nitrogen (TN) and $^{13}C/^{12}C$ ratio using a vario ISOPRIME cube (Elementar Analysesysteme GmbH, Hanau, Germany) elemental analyzer coupled to an IsoPrime100 (IsoPrime Ltd, Cheadle Hulme, UK) stabile isotope ratio mass spectrometer (EA-IRMS). Carbon isotope values are given in delta notation relative to

the Vienna Pee Dee Belemnite standard (VPDB; Hut, 1987). Data were corrected with a variety of standards from the International Atomic Energy Agency (IAEA) and in-house standards (Supplement, Table S1). Pedogenic Fe and Al fractions were analyzed by selective extractions. Oxalate extractions were conducted according to McKeague and Day (1966) by using 0.2 M ammonium oxalate (pH 3) to dissolve poorly crystalline aluminosilicates and Fe hydroxides like ferrihydrite as well as Fe and Al from organic complexes ($Fe_o$, $Al_o$). Iron present in organic complexes, poorly crystalline as well as crystalline Fe

oxides ($Fe_d$) was analyzed by extraction in dithionite-citrate following Mehra and Jackson (1960), modified by Sheldrick and



McKeague (1975). All extraction solutions were analyzed for dissolved Fe and Al by ICP-OES (Varian 725-ES, Palo Alto, California, USA).

Water-extractable OC (WEOC) was used as surrogate of DOM migrating in the soil profile (Corvasce et al., 2006). Following the procedure of Chantigny et al. (2006), 25 g of fresh, field-moist soil were extracted with 1 mM $CaCl_2$ solution at
a soil/solution ratio of 1/3. Samples were shaken horizontally for one hour at a frequency of 180 rpm at 4°C. After centrifugation for 30 min at 3,500 $g$ at 4°C, extracts were filtered through 0.45-µm cellulose-nitrate membranes (Sartorius Stedim Biotech GmbH, Göttingen, Germany). Total OC (TOC) concentrations in the extracts were measured by high-temperature combustion with a varioTOC elemental analyzer (Elementar, Hanau, Germany). The $\delta^{13}$ values of WEOC were measured with an isoTOC cube coupled to an IRMS vision (Elementar, Hanau, Germany (Leinemann et al., 2018)). The ultra
violet (UV) absorbance at 280 nm of WEOC was measured with the Specord 200 UV-vis spectrometer (Analytic Jena AG, Jena, Germany). Specific ultra violet absorbance at 280 nm (SUVA) was calculated according to Chin et al. (1994) as the ratio of UV absorbance at 280 nm and DOC concentration. Prior to fluorescence measurements, samples, if necessary, were diluted to absorbance values < 0.1 at 280 nm. Thereafter emission spectra from 300 nm to 500 nm were measured at an excitation wavelength of 254 nm (Zsolnay et al., 1999) at a Perkin Elmer LS 50 luminescence spectrometer (Perkin Elmer, Waltham,
MA, USA). For all measurements the scan rate was 100 nm min$^{-1}$ and the Ex-slit : Em-slit was 15 : 10. The stability of the instrument was checked with the Raman peak of deionized water at 350 nm. The fluorescence emission index (HIX) was calculated as the ratio of the area between 435-480 nm and the area between 300-345 nm of the emission spectrum (Zsolnay et al., 1999) using FL Winlab Software.

### 2.4 Density fractionation

Samples for density fractionation were selected in order to represent the topsoil (0-5; 5-10 cm), the upper subsoil (10-20; 20-30; 30-40; 40-50 cm), and the deeper subsoil (100-120; 120-140 cm). Density fractionation was conducted according to Golchin et al. (1994a, 1994b), with the following adjustments based on pre-tests. Aliquots of 25 ± 0.05 g bulk soil were separated into two light fractions (LF), free and occluded particulate OM (fPOM and oPOM), as well as one heavy fraction (HF) containing MAOM. After adding 125 mL sodium polytungstate (SPT) solution (SPT 0, TC-Tungsten Compounds, Grub
am Forst, Germany) with a density of 1.6 g cm$^{-3}$ (Kaiser and Guggenberger, 2007; Cerli et al., 2012), the suspensions were manually stirred and allowed to rest for one hour. Afterwards, the samples were centrifuged at 4,000 $g$ and 17°C for 30 min (Cryofuge 6000, Heraeus Holding GmbH, Hanau, Germany) and the supernatant, containing fPOM material, was filtered through 0.45-µm polyethersulfone filters (PALL Life Sciences, Ann Arbor, Michigan, USA). The fractionation of the fPOM was repeated once. In a second step, aggregates were destroyed to release oPOM by ultrasonic treatment (Sonopuls HD2200,
Bandelin electronic GmbH & Co KG, Berlin, Germany) with an energy input of 60 J mL$^{-1}$ (Gentsch et al., 2015; Schiedung et al., 2016). Prior to the treatment, ultrasonic power of the sonotrode was assessed calorimetrically and ultrasound durations were calculated according to North (1976). After centrifugation at 6,000 $g$ for 30 min, the supernatant with oPOM material was filtered as well. Both fPOM and oPOM were washed with ultra-pure water (18.2 MΩ) until the electrical conductivity of





the eluate was < 5 µS cm$^{-1}$ (Angst et al., 2016). The HF was washed three to four times with 200 mL ultra-pure water until the
conductivity was < 50 µS cm$^{-1}$. The water used for washing the HF was collected and measured for dissolved OC (C$_W$). We
also measured dissolved OC in all post-treatment SPT solutions. This SPT-mobilized C (C$_{SPT}$) was taken to represent
mobilizable and potentially labile soil OC (Gentsch et al., 2018), derived from POM and MAOM. The dissolved OC
concentrations were measured within two days after the fractionation by high temperature combustion with a limit of
quantification of 1 mg C L$^{-1}$ (Leinemann et al., 2016), using a Vario TOC cube (Elementar, Hauau, Germany). Aliquots of
both liquid phases were freeze-dried similar to the soil fractions for analysis of OC, TN, and $^{13}$C/$^{12}$C ratios by EA-IRMS. Due
to negligible amounts of POM material in the deeper subsoil samples (100-140 cm), no further differentiation between fPOM
and oPOM was done. The mean mass recovery in fPOM, oPOM; and HF after fractionation was 99.1 ± 0.9 %. The mean C
recovery after fractionation was 98.3 ± 26.5 %, including data for the mobilized C$_W$ and C$_{SPT}$. On average, 2.0 ± 2.2 % of the
recovered C was mobilized by the fractionation procedure. Nitrate and ammonia were extracted from bulk and HF samples to
analyze inorganic N contents (N$_{min}$) in order to obtain organic N (ON) contents by subtraction of N$_{min}$ from TN.  Extraction
was carried out according to Blume et al. (2010) by mixing 4 ± 0.01 g soil with 16 mL 0.0125 M CaCl$_2$ solution and shaking
the mixture for 1 h on an over-head shaker. After sedimentation, the supernatant was filtered through 0.45-µm cellulose acetate
filters (BerryTec GmbH, Grünwald, Germany) and measured by a segmented flow analyzer (San++ analyzer, Breda, The
Netherlands) with a limit of quantification of 0.1 mg N L$^{-1}$. Surfaces of the HF were further investigated by X-ray photoelectron
spectroscopy (XPS), method description and data are presented in the supplement.

## 2.5 Calculations and statistics

Soil OC stocks (kg m$^{-2}$) were calculated according to Eq. (1):

$$OC\ stock = OC \times density \times increment\ thickness \times 0.01, \tag{1}$$

with the OC content (mg g$^{-1}$) and bulk density of the fine earth fraction (g cm$^{-3}$) of each soil increment multiplied by the
increment thickness (cm). The proportion of each SOM fraction (OC$_{frac}$, in %) in percent of the total recovered OC was
calculated based on the sum of all fractions (ΣOC):

$$OC_{frac} = \frac{OC_{frac.}}{\sum OC\ (C_{fPOM},\ C_{oPOM},\ C_{MAOM},\ C_{SPT})} \times 100\ \%. \tag{2}$$

Water-extractable OC was calculated as the percentage proportion relative to OC in the respective bulk soil sample, according
to Eq. (3):

$$WEOC = \frac{OC_{extracted}}{Bulk\ OC} \times 100\ \%. \tag{3}$$





As mentioned earlier, all soil fractions released C to the $C_{SPT}$ pool, whereas the $C_W$ fraction solely originated from the MAOM in the HF fraction. Thus, the $C_W$ fraction was added to the MAOM. Further, the $\delta^{13}C$ values of the MAOM ($C_{MAOM}$, in ‰) were corrected for the $\delta^{13}C$ values of $C_W$ by using Eq. (4):

$$\delta^{13}C_{MAOM} = \frac{M_{MAOM} \times \delta^{13}C_{MAOM} + M_{Cw} \times \delta^{13}C_W}{M_{MAOM} + M_{Cw}}, \qquad (4)$$

with $M_{MAOM}$ as the C mass (mg) of the HF fraction, $M_{Cw}$ as the C mass (mg) in the total washing solution, and the $\delta^{13}C$ values (‰) of both fractions ($\delta^{13}C_{MAOM}$ and $\delta^{13}C_W$, respectively).

The $^{13}C$-labeled samples were used to calculate the proportion of native SOC ($SOC_{nat}$, in %) and label-derived SOC ($SOC_L$, in %) by Eq. (5) and Eq. (6):

$$SOC_{nat} = \frac{^{13}C_L - {}^{13}C_{in}}{^{13}C_{uL} - {}^{13}C_{in}} \times 100\ \%, \qquad (5)$$

$$SOC_L = 100 - SOC_{nat}, \qquad (6)$$

with $^{13}C_L$ as the $\delta^{13}C$ value of the labeled sample, $^{13}C_{uL}$ as the $\delta^{13}C$ value of the unlabeled control in the same soil depth, and $^{13}C_{in}$ as the $\delta^{13}C$ value of the initial labeled litter.

The recovered label-derived SOC was further quantified by estimating the SOC in each respective depth, calculating the proportion of label-derived SOC, and finally relating the label-derived SOC to the amount of the labeled C in the litter input. The total recovered label was calculated as the sum of label recovered in all OM fractions and respective soil depth increments, and given in g C m$^{-2}$. The potential loss over time was calculated as the relative decrease of recovered label in the 18 months interval between both sampling times.

If not stated differently, data are given as the mean of three replicates ± the standard deviation (SD). Depths refer to the mean depth per depth increment. $\delta^{13}C$ values (‰) of the labeled samples and fractions ($^{13}C_L$) were tested for significant enrichments compared to the natural variations of the control with the upper 90 % quantile limit of the frequency distribution (Nielsen and Wendroth, 2003), using Eq. (7):

$$^{13}C_L > \overline{X}_{uL} + \left( SD_{uL} \times t_{\Phi;p} \right), \qquad (7)$$

with $\overline{X}_{uL}$ as the mean and $SD_{uL}$ as the standard deviation of the unlabeled control samples of the respective soil increment (n = 3). The t-value originated from the Student's t-distribution ($\Phi$ = n-1, p = 0.9). Only values passing this comparison were used for recovery calculations. Data were tested for normal distribution by using Shapiro-Wilk normality test, prior to linear correlation analyses. Analyses were performed with SigmaPlot 14 (Systat Software GmbH, San Jose, USA) by using Pearson correlations (for normal distributed data, p < 0.05) or Spearman Rank Order correlations (for not normal distributed data, p <





0.05). Label recoveries in density fractions and WEOC were tested for significant changes with depth and between both sampling times by analysis of variance (ANOVA, p < 0.05) with the Tukey test as post-hoc analysis.

## 3 Results

### 3.1 Depth distribution and properties of SOC

Soil OC contents decreased strongly from about $82 \pm 57$ mg g$^{-1}$ in the topsoil to $3 \pm 1$ mg g$^{-1}$ in the upper subsoil at 50 cm soil depth (Fig. 1a). Within the deeper subsoil, OC further decreased to about 0.2 mg g$^{-1}$ in the deepest increment at 160-180 cm. Organic C stocks in the topsoil (0-10 cm depth) amounted to about 5.5 kg C m$^{-2}$ as the mean of both sampling dates, representing 48 % of the OC stock down to a soil depth of 180 cm (Table 2). Deeper subsoil only accounted for 5 % of the SOC stock (Table 2).

Directly underneath the litter layer, the majority of SOC was present as POM (Fig. 2). With increasing soil depth, the relative contribution of POM-C to SOC decreased to < 25 %, whereas the contribution of MAOM-C increased. As for SOC, also the MAOM-C content declined from about 10 to 22 mg C g HF$^{-1}$ in the topsoil to 0.3 to 0.4 mg C g HF$^{-1}$ in the deeper subsoil of 100-140 cm soil depth (Fig. 3a). The C/N ratio of the MAOM decreased with depth from about 20 in the topsoil to ~ 5 in the deep subsoil (Fig. 3b). Mean values from the first sampling in November 2016 showed minor, but consistently higher ratios compared to the second sampling in May 2018. The C$_{SPT}$ fraction amounted to 1 to 3 % of the SOC for all soil depths without a consistent trend. The contribution of WEOC showed an increase with soil depth from 0.2 % of SOC in the topsoil to 0.7 to 1.3 % in the deeper subsoil (Fig. 4a). In addition, water extracts showed a compositional change with increasing soil depth, as SUVA values decreased below 10 cm soil depth until reaching the minimum in the deep subsoil (Fig. 4b). The so called humification index derived from fluorescence spectra first increased from the topsoil to its maximum in the heavily rooted upper subsoil (Heinze et al., 2018; Wordell-Dietrich et al., 2019). Below, a constant decrease with increasing soil depth was observed (Fig. 4c).

### 3.2 Labeled litter-derived C in functional soil OM fractions

Based on $\delta^{13}$C values, bulk OM was more enriched in $^{13}$C from labeled litter than MAOM (Fig. 5a, b). Enrichments in MAOM were significant down to 20 cm soil depth compared to the control. After 40 months, the $^{13}$C enrichment of MAOM was still significant down to 20 cm, but $\delta^{13}$C values moved closer to the background (Fig. 5b). Water-extractable OC showed a significant $\delta^{13}$C enrichment to greater soil depth (60 cm) compared to the bulk soil and MAOM at both sampling dates (Fig. 5a-c). Below this depth, there was still a noticeable $\delta^{13}$C enrichment of WEOC in the labeled plots, albeit not significant.

After 22 months, about 11.2 % of the $^{13}$C-labeled litter exposed at the soil surface was recovered in the selected depth increments (0-50, 100-140 cm), whereby the contribution of the deeper depth increments was very minor (Fig. 6a). Considering the $^{13}$C of litter origin at 50-100 cm soil depth by linear interpolation between the increments 40-50 cm and 100-120 cm, this value would increase by 0.03 % only. The majority of 87 % of the recovered label was found in the first 5 cm of the topsoil.



Below 40 cm, only minor enrichments in $^{13}$C were found for individual fractions (< 0.2 % of total recovered labeled litter).

Eighteen months later, the recovered labeled $^{13}$C was lower in all depths compared to the first sampling, albeit not significant due to large variations between the plots, with a total recovery of 1.8 %. In the soil increments below 40 cm, no label was recovered at all at the second sampling in the density fractions, while minor proportions of label were still recovered within WEOC.

In total, we found that about 8.7 ± 5.6 % of the applied labeled litter was incorporated into POM in the mineral topsoil

within 22 months (Fig. 6a). This corresponds to 9.9 ± 6.1 g C m$^{-2}$ fPOM and 1.0 ± 0.9 g C m$^{-2}$ oPOM, most of it located in the 0-5 cm topsoil increment. Below, the contribution of labeled litter-derived POM decreased strongly. Nevertheless, recovered labeled litter in the oPOM fraction was detected at even greater depth (30-40 cm) after 40 months. Litter-derived $^{13}$C in the MAOM fraction represented 0.7 to 2.0 % of the summed up recovered label in the top 20 cm of the soil profile at both sampling dates (Fig. 6), a contribution of litter-derived C to the total MAOM-C of about ~ 0.2 % in the top 20 cm. Below, only smaller

contributions were found. Also the $C_{SPT}$ fraction, particularly that of the topsoil and upper subsoil of the first sampling date, showed a $^{13}$C enrichment (Fig. 6a).

However, 18 months after replacing the labeled by unlabeled litter, the proportion of labeled litter-derived C in the SPT solution decreased by 84 % on average (Table 3) and the label was only detectable down to 20 cm soil depth.

Proportions of labeled litter-derived C in WEOC illustrated clear depth and temporal trends (Fig. 7). The WEOC

fraction in the topsoil contained more than 1 % of C originally derived from the litter layer at the end of the labeling period in November 2016, with a strong decrease with depth. Below 40 cm, proportions were consistently smaller than 0.2 %. Eighteen months after litter replacement, the contribution of labeled litter-derived C in WEOC decreased to < 0.3 % in the whole soil profile.

Mean loss of the recovered litter-derived $^{13}$C over the time period of 18 months between the two samplings was 79 %,

and all fractions showed a considerable loss of > 65 % (Table 3). The losses followed the sequence: fPOM (89 %) > WEOC (80%) > oPOM (77 %) > MAOM (66 %), respectively. Similar losses were also found for the recovered material in the bulk soil with 77 % (data not shown).

## 4 Discussion

### 4.1 Particulate OM in the soil profile and contribution of litter-derived POM

Particulate OC contributed 59 ± 16 % to SOC in the Grinderwald topsoil. This high contribution of POM likely is a consequence of translocation by the meso- and macrofauna, as bioturbation can drive both, inputs and mineralization of SOC (Wilkinson et al., 2009). Results are somewhat higher than findings of Schrumpf et al. (2013) who reported 25 ± 16 % POM contribution to the SOC for several European study sites. Below the topsoil, amounts of POM were only minor (Supplement, Fig. S2). The proportional decrease of POM with soil depth confirms findings of Kaiser et al. (2002), who reported a similar

decrease in the contribution of POM to SOM from about 65 % in the topsoil to 5 % in the subsoil C horizons, illustrating a





decreasing role of root input and bioturbation in subsoil horizons (Heinze et al., 2018). Our results suggest that the majority of POM in the topsoil is not strongly connected to annual litter inputs as these are very small compared to the total POM pool. Similar to our observations, Lajtha et al. (2014b) reported for a long-term litter manipulation experiment that a 2-fold increase of litter input does not affect the C concentrations in either the bulk soil, POM, or the MAOM fraction of the mineral topsoil

and upper subsoil within 20 yrs. They concluded that forest soil C pools are not tightly coupled to changes in aboveground litter inputs on the short-term. In the upper and deeper subsoil, recent litter-derived POM was barely present after 22 months, and completely vanished after 40 months, suggesting that most POM in the subsoil rather derives from root biomass.

      In the 18 months between both samplings, we found that 89 % of recent litter-derived fPOM and 77 % of the oPOM material were lost in the soil profile. Consequently, new POM inputs are unstable and prone to decomposition, in line with

reported turnover times of < 10 years (Gaudinski et al., 2000; Baisden et al., 2002). Along with that, Crow et al. (2009) described the aboveground litter as the source of the most actively cycling soil C. The smaller C loss from oPOM compared to fPOM within 18 months (77 and 89 %) reflects a better protection of occluded POM material compared to free POM—even in this loamy sand soil (Table 1).

## 4.2 Mineral-associated OM and incorporation of litter-derived C via the DOC pathway

Beside bioturbation and rhizodeposition, translocation and sorption of DOM to the soil matrix are the other prominent processes transferring C to the subsoil (Kaiser and Kalbitz, 2012; Mikutta et al., 2019). The observed strong decrease in the contents of mineral-associated OC with soil depth (Fig. 3a) is in line with smaller root exudation rates (Tückmantel et al., 2017) and DOC fluxes (Leinemann et al., 2016) with increasing soil depth at the Grinderwald site. This also reflects a decrease of available sorption sites as sand contents are increasing (Table 1) and poorly crystalline Fe phases ($Fe_o$ contents) are

decreasing (Supplement, Table S2). Leinemann et al. (2016) observed a decrease in SUVA values of DOM with increasing soil depth, indicating a preferential sorption of plant-derived compounds in the upper parts of the soil profile. Specific UV absorbance and the fluorescence indices (HIX) of our water extracts showed a similar decline with soil depth, thus underpinning sorption as a relevant process. Decomposition of roots can substantially contribute to the subsoil SOM pool as well (Rasse et al., 2005), but since root density (Wordell-Dietrich et al., 2019) and root exudation (Tückmantel et al., 2017)

are low in the Grinderwald subsoil, we assume that the increasing share of MAOM with soil depth rather suggests an increasing importance of DOM as a dominant source of C in this forest subsoil. This depth trend was accompanied by a compositional change of MAOM as indicated by decreasing C/N ratios and increasing $\delta^{13}C$ values. Fresh litter-derived MAOM in the topsoil had typically wide C/N ratios of about 19 to 22 and low natural abundance $\delta^{13}C$ values of about -27 to -28 ‰ (Figs. 3b, 5b). Microbial processing (Six et al., 2001; Schmidt et al., 2011) and preferential sorption of $^{13}C$-depleted plant-derived phenols in

the topsoil (Guggenberger and Zech, 1994; Kaiser et al., 2001) alter the SOM characteristics with increasing soil depth by narrowing the C/N ratio and increasing the $^{13}C$ content. In line with this view, the $\delta^{13}C$ of MAOM showed a consistent increase with decreasing C/N ratio with depth (Supplement, Fig. S4), thus pointing towards an increasing contribution of microbially processed MAOM with soil depth, as proposed in the "dynamic exchange" or "cascade model" (Kaiser and Kalbitz, 2012).



Gleixner (2005) likewise attributed this trend to a higher contribution of plant and root litter in topsoil horizons, whereas the
deeper subsoil horizons are dominated by microbial-derived OM. A change towards microbial-derived OM is further supported
by decreasing SUVA and HIX values within the WEOC from the upper subsoil downwards, suggesting more aromatic and
complex plant-derived OM components like phenols being retained in the topsoil, while more microbial-derived components
like carbohydrates are present in the subsoil.

On average $1.46 \pm 0.67$ % of the fresh litter layer C was associated with minerals in the topsoil, $0.57 \pm 0.12$ % in the
upper subsoil, and only $0.01 \pm 0.02$ % in deeper subsoil compartments 22 months after adding the labeled beech litter,
emphasizing the subordinate importance of recent aboveground litter inputs to soil C stocks in all depths, especially the deeper
subsoil. Also Lajtha et al. (2014a) showed that 50 yrs of doubled litter inputs in a deciduous forest stand did not result in a net
accumulation of OC in the topsoil HF, likely as sorption sites in topsoils are already largely occupied by OM (Mikutta et al.,
2019). The element composition on the mineral surfaces of the HF supports this assumption, as the C and N contents decreased
on the mineral surfaces with increasing soil depth (Supplement, Fig. S5). Additionally, a higher content of Al and Fe within
the surface layer of soil particles with increasing depth suggests a higher proportion of uncovered mineral surfaces
(Supplement, Fig. S5). For the Dystric Cambisol under European beech, the observed recoveries of $^{13}$C in MAOM in the 22
months of labeled litter application were recalculated to average annual litter inputs from the recent litter layer into the HF of
about $0.99 \pm 0.45$ g C m$^{-2}$ yr$^{-1}$ in the topsoil, $0.37 \pm 0.10$ g C m$^{-2}$ yr$^{-1}$ in the upper subsoil, and $0.01 \pm 0.01$ g C m$^{-2}$ yr$^{-1}$ in the
deeper subsoil. Fröberg et al. (2007a) reported annual DOC fluxes of about 4-14 g C m$^{-2}$ yr$^{-1}$ in 15 cm soil depth and 1.5 to 4.5
g C m$^{-2}$ yr$^{-1}$ in 70 cm soil depth, whereof on average 14 % were derived from recent litter. This corresponds to fluxes of 0.5 to
2 g C m$^{-2}$ yr$^{-1}$ and 0.2 to 0.6 g C m$^{-2}$ yr$^{-1}$, respectively, which is similar in magnitude as the observed $^{13}$C fluxes from the labeled
litter into the HF at our study site. Given this similarity, it is reasonable to assume that recent litter-derived C contributes to
the MAOM pool in different soil depths mainly by the DOC pathway. The decreasing contribution and input rates of recent
litter-derived C further implies that there is an increasing contribution of older OC to DOC with increasing soil depth, as
likewise found when dating $^{14}$C ages of DOC (Don and Schulze, 2008).

There was a substantial decrease of the recovered $^{13}$C label in the MAOM fraction within the 18 months between the
first and second sampling. This can be explained either by desorption of litter-derived compounds (either due to microbial
degradation or abiotic exchange processes) and/or sorption of fresh unlabeled DOM, which diluted the $^{13}$C/$^{12}$C ratio to values
close to the background and thus regarded as not significantly enriched. Considering that the C content of the MAOM fraction
was rather constant at both samplings (Fig. 3a) and taking into account the considerable DOC fluxes of 0.7 to 2.1 g m$^{-2}$ yr$^{-1}$ in
the deep subsoil at 150 cm soil depth at this study site (Leinemann et al., 2016), we assume that the dominant processes
involved were the sorption of DOC from the soil solution and the accompanied replacement of litter-derived C from mineral
surfaces. In total, 1.69 g m$^{-2}$ of initially 2.54 g m$^{-2}$ recent litter-derived MAOM were lost throughout the soil profile (66 %),
most of it located in the upper 20 cm. This indicates that young OM associated with minerals, especially in the upper soil, is
not effectively stabilized by mineral surfaces (Schrumpf et al., 2013). The minor retention of $^{13}$C by soil minerals and the
subsequent remobilization of mineral-bound C in the topsoil at the Grinderwald site are both facilitated by the generally low





contents of clay (< 3 %) and pedogenic Fe and Al oxides (Supplement, Table S2), and the likely dominance of illite in the clay fraction—being a rather less sorptive phyllosilicate under acidic conditions (Kaiser et al., 1997).

Despite the fast transformation of recently formed MAOM in the topsoil, this is not resulting in a detectable significant downward translocation of C within the timeframe of 18 months. Thus, a partly downward translocation of recent litter-derived C (not labeled) within the soil solution will thus cause a continual dilution of the tracer with increasing soil depth. However, at the second sampling time, part of the translocated DOM was likely originating from horizons (O layers and upper mineral soil horizons) already enriched in $^{13}$C, thus potentially counteracting the dilution by new unlabeled DOM to a certain extent.

Microbial processing may further contribute to the observed losses, as desorbed or exchanged recent litter-derived C has a higher bioavailability (Marschner and Kalbitz, 2003).

### 4.3 Mobilizable OM – linking litter inputs and MAOM formation

The concept of C translocation from topsoil into the subsoil assumes continuous exchange processes at mineral surfaces, leading to partly desorption of microbially altered OM and thus its downward transport (Kaiser and Kalbitz, 2012). Here,

WEOC was considered to represent such mobilizable OM, being most susceptible to translocation and, hence, a source for subsoil OC stocks. Accordingly, we found an increasing importance of WEOC with increasing soil depth, as it constitutes a higher proportion to the SOC in the subsoil compared to the topsoil, implying that a higher proportion of soluble OM is present in the deeper soil compartments. A similar depth trend was detected for the mobilization of C during density fractionation, supporting the findings for WEOC. In accordance with Chantigny (2003), WEOC represented only a small part of SOC, but

was more enriched in litter-derived $^{13}$C than bulk soil SOC or MAOM (Fig. 5). Despite the higher enrichment, this accounted only for < 1.7 % of the total WEOC pool, suggesting that the majority of mobilizable OC is older than 22 months (for sampling in November 2016) or 40 months (for sampling in May 2018). In line with this, Fröberg et al. (2007b) and Hagedorn et al. (2003) reported that recent litter-derived DOC contributes only minor to the total DOC leached from the organic layer into the mineral soil.

The high δ$^{13}$C values of WEOC (Fig. 5c) and the strong decline of litter-derived C in WEOC within the upper 20 cm of the soil profile (Fig. 7) suggest that litter-derived POM is a considerable source of WEOC. For example, the beech litter residues that were removed after 22 months and sieved < 5 mm still contained up to 2 % WEOC (data not shown), which might become liberated during infiltration of soil solution. In the subsoil, WEOC likely derives to a higher proportion from MAOM next to roots and root-derived POM, the latter representing a negligible fraction in the deeper subsoil at the Grinderwald site.

In a recent soil column experiment, Leinemann et al. (2018) showed that 20 % of the MAOM can be replaced by percolating DOM in samples collected from three depths down to 100 cm soil depth. Most intriguing, we did not observe a downward migration of the $^{13}$C label within WEOC 18 months later, again pointing to losses of litter-derived C in all soil increments by microbial decomposition. This assumption is supported by findings from Tipping et al. (2012) who showed that the majority of DOM released from the mineral matrix can be lost by mineralization. This also matches well to the fact that subsoil MAOM

is only to a minor extent derived from recent litter-derived C sources. In summary, topsoil WEOC at least partly derives from





the recent litter layer, whereas this is not the case in the deeper soil. This finding thus supports the view, as proposed in the cascade model, that the downward migration of C involves the mobilization of older SOM components.

## 5 Implications

A prominent concept for the build-up of soil OC stocks not only considers the input of plant residues into soil but also the
subsequent fate of OM inputs in the soil, where C is assumed to undergo a sequence of cycles including sorptive retention, microbial processing, and desorption on its way down the soil profile (Kaiser and Kalbitz, 2012). This study thus investigated the impact of recent aboveground litter for OC sequestration and the subsequent partitioning of litter-derived C in different soil layers and OM fractions. Annual C inputs from the recent litter layer into the mineral soil were relatively low compared to the C already stored in soil. Most of new litter-derived C is retained in the topsoil, mainly as POM. In fact, we did not find
a translocation of considerable amounts of recent litter-derived C into the deep subsoil, indicating that most translocated OM at the study site is of older age. Our field study supports the concept that C accumulation in deeper soil involves several (re)mobilization cycles of OM during its downward migration. The large C losses in the topsoil during a period of 18 months without concomitant increase in subsoil C indicate that the young SOC, especially in the form of POM, represents an actively cycling C pool. Slowest turnover of litter-derived C was observed for MAOM compared to both POM fractions, supporting
the assumption that accessibility and sorptive stabilization reduces the vulnerability of OM to microbial decomposition. The loss of about 66 % of the C from the HF within 18 months, however, confirms earlier findings (Schrumpf et al., 2013) that part of the MAOM is rather labile, especially in the presence of less reactive minerals such as quartz or illite at our study site.

In summary, given the highly active C cycling in the topsoil and upper subsoil at the Grinderwald site, only marginally C from a recent litter layer enters the deep mineral subsoil. The build-up of subsoil C stocks is thus not connected to a direct
transfer from the litter layer but goes along with repeated sorption and remobilization cycles of OM during downward migration over a much longer period than 3.5 yrs.

*Author contribution.* AD, KK, RM, and GG designed the experiment and PL, FK, and PWD carried it out in the field. PWD,
LRD, FK, and PL processed the samples and did the analyses. SKW conducted the XPS measurements. PL took the lead in preparing the manuscript, with contributions from all co-authors.

*Competing interests.* The authors declare that they have no conflict of interest.




*Acknowledgements.* This study was funded by the Deutsche Forschungsgemeinschaft (DFG) within the research unit SUBSOM (FOR1806) – "The Forgotten Part of Carbon Cycling: Organic Matter Storage and Turnover in Subsoils". The authors would like to thank Timo Leinemann for all his work in the first SUBSOM phase, including the labeling and sampling for this experiment. We thank Frank Hegewald for support in the field, Manuela Unger for carrying out the $^{13}$C analysis of the water extracts, and all lab teams for assistance in the lab. We also want to thank Leopold Sauheitl for helpful discussions about density fractionation and detection limits and Jürgen Böttcher for helpful discussions about data analysis. We further thank Markus Koch and Moritz Rahlfs for valuable comments on earlier versions of this manuscript.

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



**Table 1.** Selected soil properties given as the mean of all three sites (n = 3) and standard deviation in brackets (data adopted from Leinemann et al. (2016)).

| Horizon WRB[1]/ KA5[2] | Depth [cm] | OC[3] [mg g⁻¹] | TN[4] [mg g⁻¹] | pH [CaCl₂] | Clay ---------- | Silt [mg g⁻¹] | Sand ---------- |
|---|---|---|---|---|---|---|---|
| AE/Ahe | 0-10 | 15.18 (1.72) | 0.59 (0.06) | 3.2 (0.2) | 19 (3) | 282 (56) | 699 (57) |
| Bsw/Bsv | 10-23 | 9.59 (2.52) | 0.41 (0.09) | 3.5 (0.4) | 27 (11) | 307 (81) | 666 (90) |
| Bw/Bv | 23-67 | 4.65 (1.96) | 0.26 (0.04) | 3.9 (0.1) | 26 (4) | 332 (99) | 642 (103) |
| C/Cv | 67-99 | 1.07 (0.46) | 0.08 (0.02) | 3.9 (0.2) | 29 (8) | 255 (41) | 716 (47) |
| 2C/IICv | 99-138 | 0.34 (0.11) | 0.07 (0.09) | 4.1 (0.1) | 21 (14) | 87 (55) | 891 (66) |
| 3C/IIICv | 138-175 | 1.05 (0.11) | 0.10 (0.11) | 4.0 (0.3) | 32 (44) | 268 (422) | 700 (466) |
| 4C/IVCv | 175+ | 0.29 (0.14) | 0.03 (0.04) | 3.9 (0.2) | 19 (6) | 58 (8) | 923 (14) |

[1] according to IUSS Working Group WRB (2014)

[2] according to Ad-hoc-Arbeitsgruppe Boden (2005) , i.e. according to German soil classification

[3] Organic carbon (OC)

[4] Total nitrogen (TN)






**Table 2**. Mean OC stocks in bulk soil of different soil compartments down to 180 cm presented as absolute values and as percent of total soil OC stock (n = 12 and standard deviation is given in brackets).

| Soil compartment | Depth [cm] | Mean C stock [kg m$^{-2}$] | % of total C stock |
|:---:|:---:|:---:|:---:|
| Topsoil | 0-10 | 5.51 (3.67) | 48 (13) |
| Upper subsoil | 10-50 | 3.91 (0.67) | 40 (10) |
| Mid subsoil | 50-100 | 0.76 (0.35) | 7 (3) |
| Deeper subsoil | 100-180 | 0.50 (0.33) | 5 (3) |



**Table 3**. Mean contents of litter-derived OM in different soil fractions of all depth increments used for density fractionation (0-50 cm; 100-140 cm) for the sampling 22 months after labeled litter application in November 2016 and the sampling 40 months after labeled litter application in May 2018 (n = 3 and standard deviation is given in brackets) and the calculated loss of litter-derived OM between both samplings (18 months) in percent.

| | Recovered November 2016 [g m$^{-2}$] | Recovered May 2018 [g m$^{-2}$] | Loss over time [%] |
|---|---|---|---|
| MAOM[1] | 2.54 (0.92) | 0.85 (0.52) | 66 |
| fPOM[2] | 9.89 (6.14) | 1.11 (0.96) | 89 |
| oPOM[3] | 0.98 (0.91) | 0.23 (0.24) | 77 |
| C$_{SPT}$[4] | 0.54 (0.35) | 0.08 (0.08) | 84 |
| WEOC[5] | 0.15 (0.06) | 0.03 (0.01) | 80 |

[1] Mineral-associated organic matter (MAOM)

[2] Free particulate organic matter (fPOM)

[3] Occluded particulate organic matter (oPOM)

[4] Sodium polytungstate mobilizable C (C$_{SPT}$)

[5] Water-extractable organic C (WEOC)


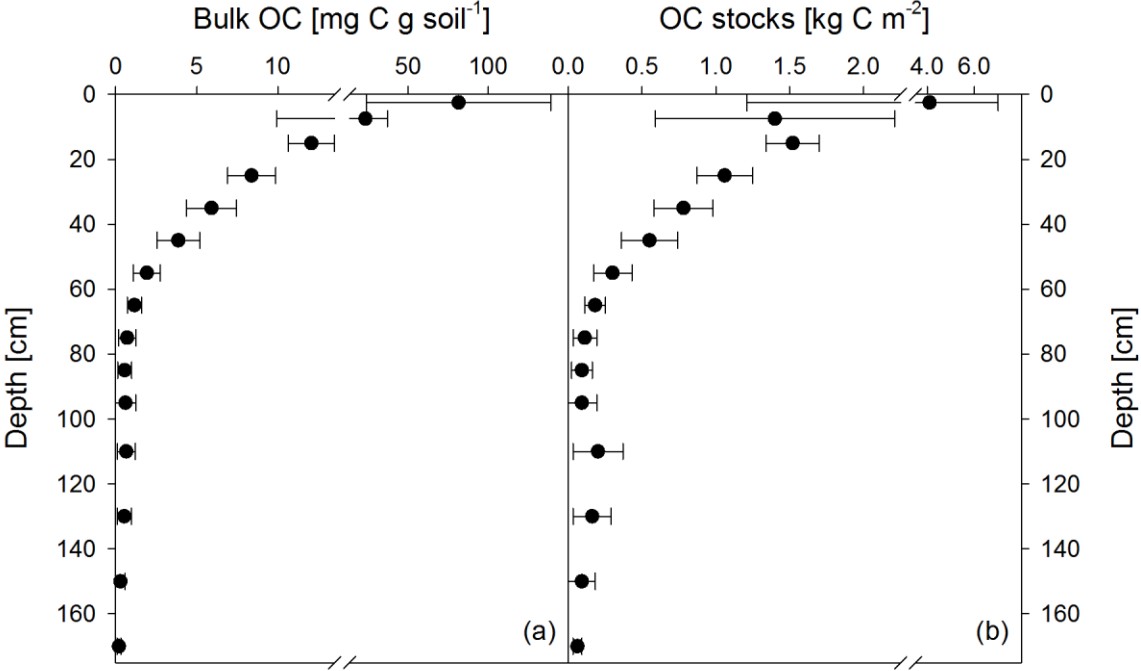

**Figure 1**. Mean bulk OC contents of both sampling times (November 2016, May 2018) (a) and calculated carbon stocks as the mean of both sampling times (b). Apparent re-increasing OC stocks below 100 cm are the result of doubling the thickness of the analyzed depth increments (i.e. 5 cm increments from 0 to 10 cm, 10 cm increments from 10 to 100 cm, and 20 cm increments from 100 to 180 cm). Data show the mean of 12 samples and error bars show the standard deviation.




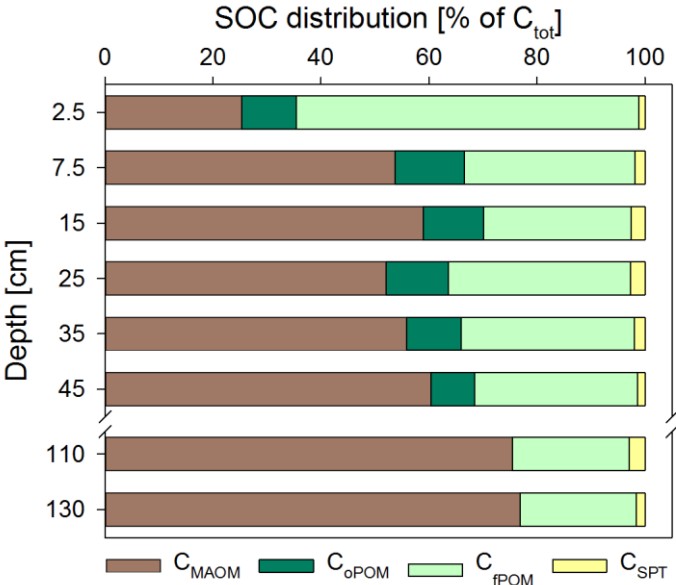

**Figure 2**. Soil organic carbon distribution separated into the mineral-associated organic matter (MAOM) within the heavy fraction corrected for the loss during washing by the $C_W$ contents ($C_{MAOM}$), the occluded particulate organic matter (oPOM) fraction ($C_{oPOM}$), the free particulate organic matter (fPOM) fraction ($C_{fPOM}$), and the carbon found in the density solution ($C_{SPT}$) as the mean of both samplings (n = 12, standard deviation varied for $C_{MAOM}$ between 7-19 %, for $C_{oPOM}$ between 2-5 %, for $C_{fPOM}$ between 7-19 %, and for $C_{SPT}$ between 0.3-5 %).



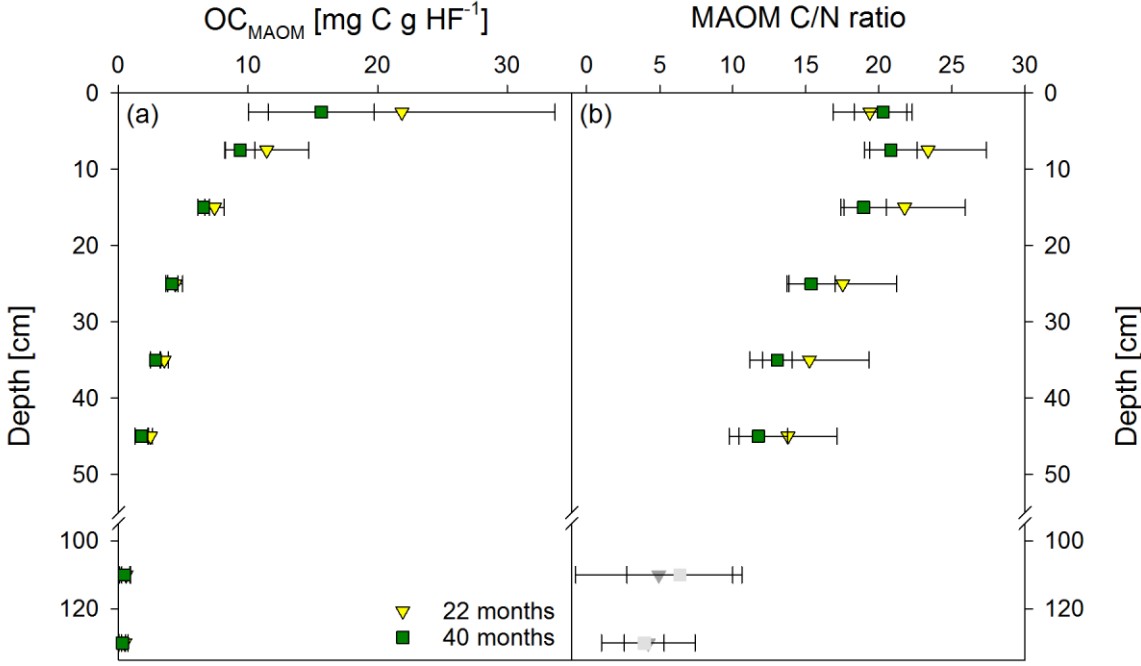

**Figure 3**. Mean organic C contents in the heavy fraction (HF) (a) and mean C/N ratios (b) of the mineral-associated organic

matter fraction (MAOM) from both sampling times, 22 months and 40 months after labeled litter application (n = 6, error bars

represent the standard deviation). Nitrogen contents in samples below 100 cm were not reliable and C/N ratios are therefore

marked in grey.



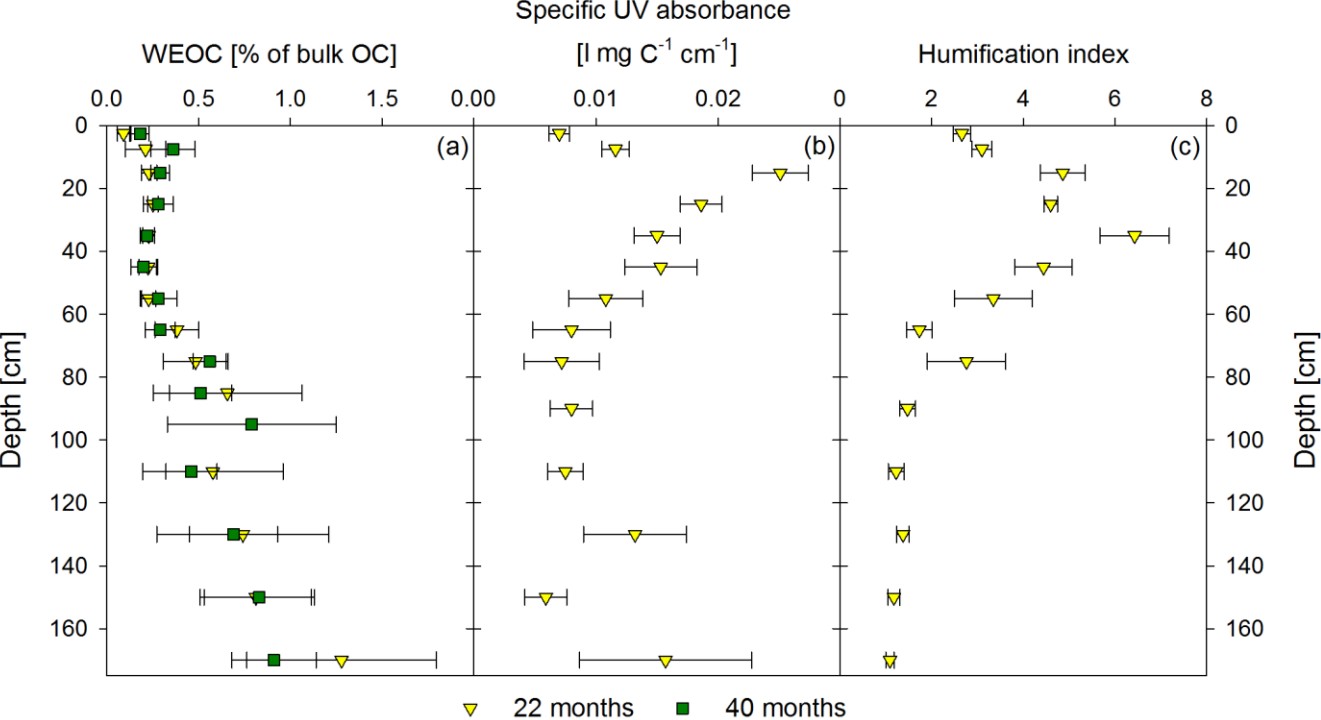

**Figure 4**. Mean proportion of water-extractable organic C (WEOC) per depth increment in % of the total soil organic C in bulk soil for both sampling times, 22 months and 40 months after labeled litter application (n = 6, error bars represent the standard deviation) is shown in (a). Specific UV absorbance at 280 nm (b) and humification index deduced from fluorescence spectra (c) of the water extracts are given as the mean (n = 6) of the first sampling in November 2016. Error bars represent the standard error.

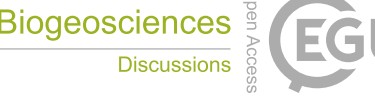

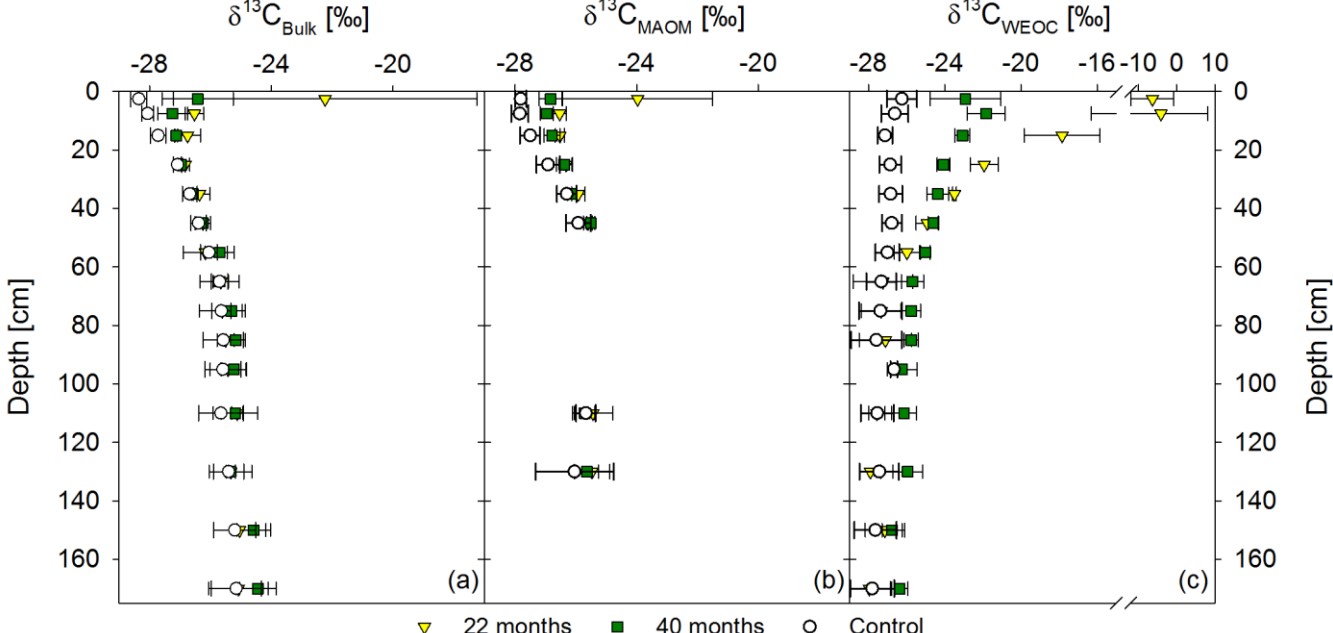

**Figure 5**. Mean δ¹³C values in the bulk soil (a), the mineral-associated (MAOM) fraction (b), and the water-extractable (WEOC) fraction (c). The graphs show labeled samples of both sampling times, 22 months and 40 months after labeled litter application in colored symbols, compared to the respective unlabeled background distribution in white symbols. Labeled samples represent the mean of three replicates per sampling time, while the control represents the mean of both sampling times (n = 6). Please note that the X-axis in (c) has a different scale.





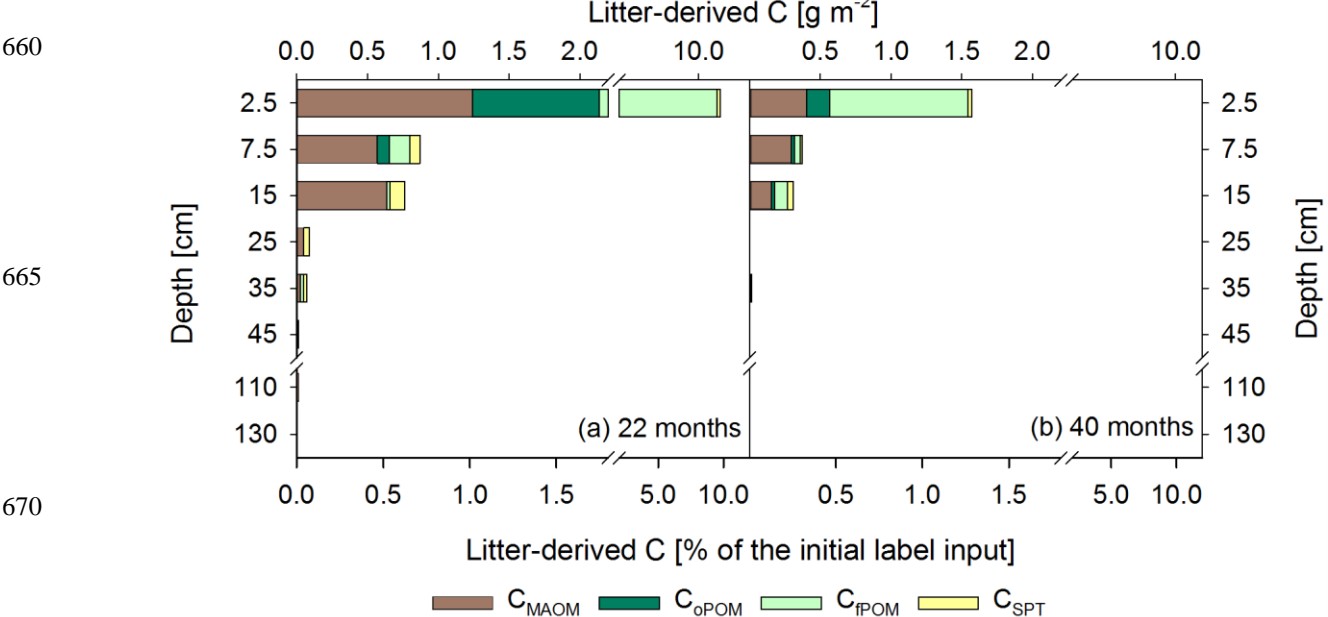


**Figure 6**. Mean labeled litter-derived $^{13}C$ in different soil fractions (mineral associated organic matter (MAOM), occluded

particulate organic matter (oPOM), free particulate organic matter (fPOM), and SPT mobilizable ($C_{SPT}$)) in g m$^{-2}$ on the upper

X-axis and in % of the label added with the replaced litter on the lower X-axis at sampling in November 2016, 22 months after

labeled litter application (a), and May 2018, 40 months after labeled litter application (b). Bars show the sum of all fractions

per depth increment, while the different colors represent the respective contribution of each fraction to the total recovery (n =

3). According to ANOVA tests there were no significant changes in $^{13}C$ recovery for each fraction with depth per sampling,

due to high standard deviations in the range of 0.02–0.53 for $C_{MAOM}$, 0.01–0.75 for $C_{oPOM}$, 0.02–4.9 for $C_{fPOM}$, and 0.01–0.13

for $C_{SPT}$.



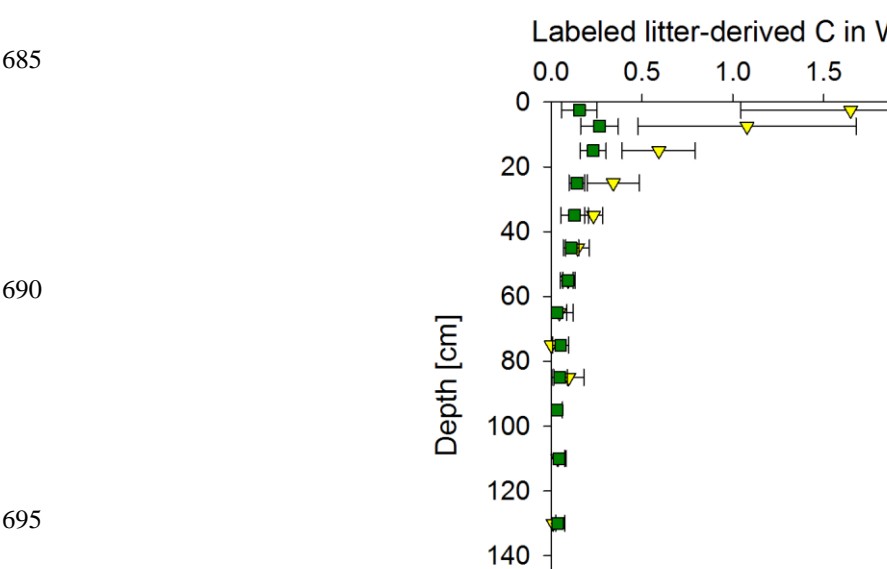

**Figure 7**. Mean proportion of litter-derived C in water extracts (WEOC) in % of the initial label input for both sampling times, 22 months and 40 months after labeled litter application, with soil depth (n = 3; error bars represent the standard deviation). According to ANOVA tests significant changes between both samplings were only present in the 0-5 cm and 10-20 cm increments ($p < 0.05$). Significant differences between soil increments were only present for the topsoil increments compared to all subsoil increments for each sampling time.