# Peer review of "Relevance of aboveground litter for soil organic matter formation – a soil profile perspective"

_Biogeosciences, 2019_

## Short Comment (SC1) · 8 Jan 2020

The following 5 papers report on field studies of enriched background 14C isotopic tracers for multiyear controlled litter additions and the transfer of those labels into the soil. You might modify your statement on line 65 to recognize these efforts. The Kramer et al. 2010 paper is probably the most relevant, and you have already cited the related mesocosm study (Froberg et al. 2009)

1. Tipping E, Chamberlain PM, Fröberg M, Hanson PJ, Jardine PM (2012) Simulation of carbon cycling, including dissolved organic carbon transport, in forest soil locally enriched with 14C. Biogeochemistry 108:91-107, doi 10.1007/s10533-011-9575-1. 2.

Parton WJ, Hanson PJ, Swanston C, Torn M, Trumbore SE, Riley W, Kelly R (2010) ForCent model development and testing using the Enriched Background Isotope Study (EBIS) experiment. JGR-Biogeosciences 115:G04001, doi:10.1029/2009JG001193 3. Kramer C, Trumbore S, Fröberg M, Cisneros-Dozal LM, Zhang D, Xu X, Santos G, Hanson PJ (2010) Recent (<4 year old) leaf litter is not a major source of microbial carbon in a temperate forest mineral soil. Soil Biology and Biochemistry 42:1028-1037. 4. Riley WJ, Gaudinski JB, Torn MS, Joslin JD, Hanson PJ (2009) Fine-root mortality rates in a temperate forest: estimates using radiocarbon data and numerical modeling. New Phytologist 184:387-398. 5. Fröberg M, Hanson PJ, Trumbore SE, Swanston CW, Todd DE (2009) Flux of carbon from 14C-enriched leaf litter throughout a forest soil mesocosm. Geoderma 149:181-188. [Mesocosm study in support of the larger field EBIS effort.]

---

## Referee Comment (RC1) · Anonymous Referee #3 · 10 Jan 2020

This study investigated the impact of aboveground litter for soil organic carbon (C) sequestration and the subsequent partitioning of litter-derived C in different soil layers and OM fractions. In general, I think the data are solid and the results are valuable for understanding fates of litter C input. I have some minor comments/suggestions that could improve the manuscript.

Lines 37 and 38: This statement may be correct only for natural ecosystems. For example, OM disturbance due to tillage may be a pathway for cropping systems.

Line 99: Did you also observe the amount and chemical properties of the litter? These factors could impact litter decomposition and are important for interpreting the results.

Line 271: "...both, inputs...". Line 301: For "DOM", did you mean DOM leached from surface soil layers? Is it possible that rhizodeposition still made a considerable contribution in subsoil MAOM although root density and exudation were low, given that subsoil MAOM contents were also very low? In addition, it looks that microbial decomposition of root derived C may also increases 13C values and decrease C/N ratios of MAOM; so I am wondering if the observations can fully support the conclusion that DOM leached from the surface soil layers was a dominant source. Lines 360 to 364: Could you explain where the majority of litter-derived C goes; emitted as CO2? If so, why the older mobilizable OC did not emit as CO2? Line 366: Did you measure the amount of litter residues after 22 months? Line 395: This statement (and may be statements in other places) is also related to the comment on root-derived C contribution to subsoil OM. Figure 1: What about the differences of bulk OC between these two sampling times; increasing, decreasing, or no detectable change? Figure 3: I would suggest deleting the grey points if they were not reliable.

---

## Referee Comment (RC2) · Anonymous Referee #2 · 15 Jan 2020

This is an interesting analysis estimating the contribution of leaf litter on soil organic matter formation of each soil layers. Generally, this is a well performed field study on a relevant subject. The manuscript is quite interesting and decently written, although some descriptions and conclusions are inaccurate. I suggest revisions to address some of the issues I raise below.

Lines 23-26: This description is inaccurate. 0-10-cm soil sequestrated 0.99 g C m-2 yr-1 from labeled litter, 0.37 g C m-2 yr-1 in the 10-50-cm soil layers. It is not surprising, compared to the considerably large contribution of 0-10 cm soil C pools. 48% of the SOC stocks (0-180 cm) were sequestrated in the top 10 cm soil layer (Table 2).

Lines 34-36: Most studies focused on SOC dynamics only in 0-10 cm soil layers? The concepts of "topsoil" and "subsoil" are confusing throughout the text. According to my understanding, the authors described the soils in the 10-to-180-cm layers as "subsoil" involving their own results. But the topsoil described here is obviously not 0-10 cm only. Lines 38-41: This statement is not correct. Below-ground inputs may more important contribution than the litter for SOC accumulation (Nadelhoffer and Raich, 1992; Majdi, 2001; Pausch and Kuzyakov, 2018). Nadelhoffer, K. J., and Raich, J. W.: Fine root production estimates and belowground carbon allocation in forest ecosystems. Ecology, 73, 1139–1147, 1992. Majdi, H.: Changes in fine root production and longevity in relation to water and nutrient availability in a Norway spruce stand in northern Sweden. Tree Physiol., 21, 1057–1061, 2001. Pausch, J., and Kuzyakov, Y.: Carbon input by roots into the soil: quantification of rhizodeposition from root to ecosystem scale. Glob. Change Biol., 24, 1-12, 2018. Lines 64-66: I noticed and agreed with the comments from Paul Hanson. And: Guelland K, Esperschütz J, Bornhauser D, et al. Mineralisation and leaching of C from 13C labelled plant litter along an initial soil chronosequence of a glacier forefield. Soil Biology and Biochemistry, 2013, 57: 237-247. Kammer A, Schmidt M W I, Hagedorn F. Decomposition pathways of 13 C-depleted leaf litter in forest soils of the Swiss Jura. Biogeochemistry, 2012, 108: 395-411. Line 116: It's important to measure the mass of litter (both for initial and after 22-months) for estimating the relative contribution of the sequestrated C from litter? This is my primary concern. Line 217: SOC content in 0-10 cm soil (8.2% here) is largely different (> 5 times) from that given in Table 1 (1.5%, the same forest plot or stand, their previous study). Is there any special on the location of the soil sampling in this study?

---

## Referee Comment (RC3) · Anonymous Referee #1 · 19 Jan 2020

This is a nice straightforward presentation of a field study investigating the fate of surficial litter-derived carbon as it enters and travels down the soil profile. The introduction presents a good overview of the current scientific understanding and of the study objectives. As mentioned by previous reviewers, it may benefit from acknowledging past studies using radioactive carbon, as well as the few studies using stable carbon to follow the fate of surface litter. The methodological approach is described in sufficient detail, and the results are concisely presented (thank you!). This paper presents a case study-results from a specific soil. There is still value in getting the work published as is, as I agree with the authors that quantitative information on the fate of carbon inputs after they enter soils is still mostly missing. Out of curiosity, why was that particular study

site chosen? For convenience, or was there another more scientific reason? However, I have been trying to wrap my head around the potential broader significance of the presented study. The studied site seems to be affected (to a large extent?) by bioturbation, and a lot of recent carbon was recovered in particulate organic matter. How would the situation be different in the case of soils less affected by soil fauna? Not only in term of the topsoil carbon, but also more importantly in term of DOC leaching and redistribution lower in the profile? Would fluxes then be more important? Lastly, how would the results look like if the study had been conducted longer? Eighteen months may not be enough time to see redistribution at depth.

---

## Referee Comment (RC4) · Anonymous Referee #4 · 20 Jan 2020

This manuscript is well written and organised. At this stage I have comments/concerns regarding some methodological aspects which are elaborated below. Also, the points touched on in the discussion are clear, but don't bring the arguments back (explicitly ) to the hypotheses presented in the Introduction.

-How was the highly labelled litter (i.e. the source of the enriched C), produced? Is it homogeneously labelled? This is important because if labelling is not homogeneous only some compound types and pools of C will be traceable, which may not represent the whole plant C well, or bias it against the movement and stabilization of certain litter-derived compounds. This could lead to substantial underestimation/over estimation (?)

[Figure]

of the contributions or surface litter. It would also affect the overall estimation of loss. Given the type of goal, which is mainly one of quantifying contribution (vs. comparing different treatments) this is of high importance and is potentially concerning.

-Question: how were the labelled and unlabelled litter mixed?

-I am assuming that the natural litter added is "fresh" litter? Or at what stage of decay is? And what about the labelled litter? Is it seneced? Fresh? If the two litters were are different stages, this would have implications, because of the differential composition of C pools, depending on the potential scenarios. In this case, what could they be? This is also a consideration for the initial mixed litter.

- I don't understand how the 20mm (do you mean 2 cm mesh?) could prevent the leaching of the naturally fallen litter to reach the soil

- I am confused by the handling of the samples for water extractions. Line 115 says they were soil subsamples were frozen directly after sampling for water extractions, but later one it says field-fresh samples were extracted. The freezing and thawing will have an impact on the C composition of the soil solution from the breaking of the microbial cells, putting cellular contents into solution, potentially. Then also, if the soils were not extracted soon after field collection the C composition of the soil solution and its isotopic composition would potentially change too. With such the low levels of enrichment that reach the sub-soil, these unintended impacts of the handling could alter the results.

-It would be good to explain the general purpose of the investigation of HF surfaces in the methods.

-Not methodological: In the Introduction, some potential reasons for the 13C enrichment with depth are mentioned; there are some new developments about this gradient such as evidence also suggesting there is a contribution also of the microbial composition of the necromass (e.g. Biogeochemistry 2015, 124: 13-26)

---

## Referee Comment (RC5) · Anonymous Referee #5 · 22 Jan 2020

The article deals with a significant question that is the redistribution of fresh litter carbon into different soil C pools and the processes involved in the transfer and turnover of carbon in the whole soil profile. The use of stable isotope labeling in a field study seems really adapted. However some information is missing, sometimes with significant importance for interpretation. Main concerns: o What is the percentage of remaining litter (in mass and labelling) after 22 months? This data is important to assess the percentage of litter lost by mineralization compared to the part that did not enter the soil, and to know if the incorporation of litter fits well with natural conditions (11% seems low). o What are the properties of labeled and unlabeled litter? Was the labeling realized through continuous or pulse technique? In relation to this question, what is the $\delta$13C of

the remaining litter after 22 months (is it consistent with the $\delta$13C measured for initial litter or may have the incorporation process been discriminant?). The interpretation of the labeling calculation could be different if the labeling is not homogenous. o All the data that would allow comparing the properties of the different plots, especially labeled/unlabeled plots. For example, authors always averaged the control and the labeled plots for soil properties: the C contents and stocks (figure 1), the distribution in density fractions (in figures 2 and 6, the variability is high as mentioned in the caption: is it distributed randomly between labeled and unlabeled plots or is there significant differences?), the C/N (figure 3), WEOC (figure 4). If significant differences exist between the labeled/unlabeled plots, the interpretation of the low difference in isotopic signature (figure 5) could be limited. o Table S1: add the value of the reference (C, N or $\delta$13C). Moreover, was a labeled standard (in-house) used since the initial enrichment applied was high (1241-1880 prmil)? What is the maximum $\delta$13C measured in the soil fractions and used for calculation? Minor concerns: - A 2-mm mesh was used to prevent new litter input during first 22 months: what about a potential leaching of additional unlabeled litter? As mentioned lines 366-368: WEOC release is possible. - Was the WEOC extracted on frozen samples (line 115) or on field-moist samples (line 129)? - Line 322-325: the sentence is not clear for me. Which recalculation was done? Is it to correct the input of litter to the soil of the experiment that was not representative of typical "annual" litter input? If it is the case: what about this difference (line 98-99, authors mentioned a "equivalent amount of litter" added to the plots)? - The XPS part does not seem to be related to the study. Is it necessary? Additional Table S3, Figure S1, figure S3 and figure S6 are not cited in the text for example. At least Table S3 and Figure S5 seem redundant. I would delete the Table S3 (or simplify it? The replicates should be averaged). - Line 131: Were the filters pre-rinsed? Was the effect of cellulose pollution on $\delta$13C of WEOC assessed? Is it negligible (depending on the DOC content of the WEOC extract)? If this pollution is equivalent for the labeled/unlabeled WEOC (and if WEOC is equivalent for labeled/unlabeled plots), it may not impact the isotopic calculation but should be mentioned. - The WEOC extraction and the density

fractionation were done in parallel (not sequential). So is there a relation between the DOC collected in SPT fraction and the WEOC? Line 227, authors mentioned that the "no consistent trend" was observed for DOC of SPT. What about the $\delta$13C of SPT? - Line 169-174: Measurement of nitrate and ammonium, for calculation of organic nitrogen were mentioned, but never used in result. Delete? - Line 174-175: Is the XPS really useful for this study? - Line 228: cite Figure 2 - Line 248: cite Figure 5c - Line 258: cite Figure 6b - Line 267: Sentence is not clear: what means "similar loss for recovered material", is it 77% of mass or of carbon? - Table 3: add the % of initial litter that was lost by mineralization compared to the material remaining after 22 months. Express the values in % of C "entering" the soil. - Figure 1, 3, 4: show the mean and SD of labeled and unlabeled plot, of 22 and 40 months.

---

## Author Comment (AC2) · 5 Mar 2020

RC1 Comments by Referee #3, 10.01.2020

Introduction

This study investigated the impact of aboveground litter for soil organic carbon (C) sequestration and the subsequent partitioning of litter-derived C in different soil layers and OM fractions. In general, I think the data are solid and the results are valuable for understanding fates of litter C input. I have some minor comments/suggestions that could improve the manuscript.

**Authors response**

We want to thank the referee for his positive feedback and the helpful and constructive criticism, which helped to improve the manuscript.

**1. Comment**

Lines 37 and 38: This statement may be correct only for natural ecosystems. For example, OM disturbance due to tillage may be a pathway for cropping systems.

**Author response**

We agree with the Referees comment and modified the sentence in line 37 to 38 of the original manuscript as follows: "In forest ecosystems, major pathways of OM to enter subsoils are rhizodepositions, root exudation and dissolved organic matter (DOM) leached from the horizons above (Wilkinson et al., 2009; Rumpel and Kögel-Knabner, 2011; Kaiser and Kalbitz, 2012)."

**2. Comment**

Line 99: Did you also observe the amount and chemical properties of the litter? These factors could impact litter decomposition and are important for interpreting the results.

**Author response**

The amount of litter was about 275 g m-2. This was defined according to measurements of Meier et al. (2005), who reported a litter input of a beech forest at 165 of 427 g m-2. Chemical properties were not analyzed, but since the labeled leaves were harvested from the same tree type (Fagus sylvatica) as in the research forest, we assume that the chemical properties of the litter resemble the natural environment of our study site. We recognize this comment and modified the sentence in line 98 to 99 of the original manuscript as follows: "For the labeling, the natural litter layer was removed manually and replaced by an equivalent amount of 275 g 13C enriched beech litter per m-2, representing a typical input of beech litter in Germany (Meier et al., 2005).
Labeled litter was prepared as ...."

3. Comment

Line 271: "...both, inputs...".

Author response

We are unsure about the intention of this comment, but we assume it aimed at the comma? But manuscript text and comment are the same. No changes to the sentence in the original manuscript were made.

4. Comment

Line 301: For "DOM", did you mean DOM leached from surface soil layers?

**Author response**

In the first passage of this sub-chapter 4.2 (from line 290 to 314), we discuss the overall role of DOM for MAOM formation, without a specific focus on litter-derived DOM but rather relate the DOM in general, i.e. of different source. To prevent possible misunderstandings, the sentence in the original manuscript was modified in the following way: "Decomposition of roots can substantially contribute to the subsoil SOM pool as well (Rasse et al., 2005), but since root density (Heinze et al., 2018; Wordell-Dietrich et al., 2019) and root exudation (Tückmantel et al., 2017) are low in the Grinderwald subsoil, we assume that the increasing share of MAOM with soil depth rather suggests an increasing importance of DOM as a dominant source of C in this forest subsoil, irrespective of its origin."

**5. Comment**

Is it possible that rhizodeposition still made a considerable contribution in subsoil MAOM although root density and exudation were low, given that subsoil MAOM contents were also very low?
**Author response**

A considerable contribution of rhizodepositions is possible, as we also found that about 20 % of the deep subsoil SOC is present as POM (Fig. 2), most likely derived from roots, but we assume that it is not the dominant source of subsoil MAOM, as we discussed this in lines 298 to 301 of the original manuscript. And, yes, in absolute number, the content of subsoil MAOM is very low (Fig. 3a). In this paragraph, we wanted to highlight the shift in the importance of the different functional OM fractions from 25 % to 77 % with increasing soil depth.

**6. Comment**

In addition, it looks that microbial decomposition of root derived C may also increases 13C values and decrease C/N ratios of MAOM; so I am wondering if the observations can fully support the conclusion that DOM leached from the surface soil layers was a dominant source.

**Author response**

We agree that microbial decomposition of either root-derived C or also litter-derived C may increase 13C values. Together with preferential sorption of 13C-depleted substances, both processes account for the 13C pattern with soil depth, as we discuss in line 304 to 306. Since we do not link our observations and conclusions to recent litter-derived C alone in lines 290 to 314, we think that the response given in Referee's comment #4 and the respective modification in the text are sufficient to clarify that we relate this observation to DOM of different origin. We want to add here that the natural 13C pattern with soil depth was taken into account and used to determine significant enrichments of the labeled samples. This was expressed in eq. 7 (lines 204 to 210).

**7. Comment**

Lines 360 to 364: Could you explain where the majority of litter-derived C goes; emitted as CO2?
**Author response**

This is a good and very important comment/question, which we will definitely address. We made a detailed mass balance regarding the fate of recent litter layer-C, including DOC monitoring and surface CO2 monitoring. Both will be subject of another publication (currently in preparation), which will focus on the budget in contrast to the present publication where focus in on the fate of litter-derived OM in soil. To answer the question: The majority of the labeled litter-C on the one hand indeed emitted as CO2 ( $\sim$  36-40 %) and on the other hand remained in the litter layer ( $\sim$  35-40 %).

**8. Comment**

If so, why the older mobilizable OC did not emit as CO2?

**Author response**

Older OC definitely did emit as CO2 (katabolic pathway), or is recycled by soil microorganisms and consequently used as a source to build-up biomass (anabolic pathway). Microbial decomposition is also the primary reason for the strong decrease of SOC (Fig. 1) and MAOM-C (Fig. 3) with increasing soil depth. What we wanted to highlight in the lines 360 to 364 was, that the mobilizable OC fraction contains predominantly C older than 22 months, despite showing a higher 13C value compared to bulk soil or MAOM.

**9. Comment**

Line 366: Did you measure the amount of litter residues after 22 months?

**Author response**

Yes, the removed litter residues after 22 months were measured and amounted to 405 g m-2 per site. Considering an initial mass of 275 g m-2 added litter, we removed about 130 g m-2 more litter than applied. This difference may have resulted from freshly fallen litter material, which was smaller than the mesh size and therefore accumulated during
the 22 months. The proportion of remaining labeled litter within the removed litter was about 25 %, corresponding to about 35-40 % of the initial applied labeled litter, mentioned in comment #7. We modified the sentence in line 102 to 103 in the original manuscript as follows to include this information: "In November 2016, the labeled litter was removed manually and amounted to an average of about 405 g m-2 per plot. We thus removed more litter than we initially applied due to incorporation of small leaf debris and beechnut shells during the 22 months. About 25 % of the removed litter were residues of the applied labeled litter."

**10. Comment**

Line 395: This statement (and may be statements in other places) is also related to the comment on root-derived C contribution to subsoil OM.

**Author response**

We agree that roots should always be considered in the context of soil OM. But since we already discussed the (not dominant) impact of roots at our study site in section 4.2, we prefer to end the manuscript with highlighting our main implication and not with findings of other publications (e.g. Rasse et al. 2005).

Rasse, D. P., Rumpel, C. and Dignac, M.-F.: Is soil carbon mostly root carbon? Mechanisms for a specific stabilisation, Plant Soil, 269(1–2), 341–356, doi:10.1007/s11104-545 004-0907-y, 2005.

**11. Comment**

Figure 1: What about the differences of bulk OC between these two sampling times; increasing, decreasing, or no detectable change?

**Author response**

For the majority of depth increments (9 out of 14), there were no changes between both sampling times. However, for 5 increments, including 0-5 and 5-10 cm, bulk soil
OC was smaller at the second sampling compared to the first, likely due to variations in litterfall, bioturbation, and decomposition as a result of differences in precipitation. In the 22 months of litter application, about 950 mm precipitation was measured while it was only about 570 mm in the 18 months thereafter.

12. Comment

Figure 3: I would suggest deleting the grey points if they were not reliable.

**Author response**

We highly discussed this topic among the authors before the submission and now during the review process. We know that such low values are not realistic for SOC in any soil depth. The reason for these values is a nitrogen content close to the detection limit. Nevertheless we included the values, since all other figures show complete data sets and we wanted to be consistent and also transparent by not excluding data. To prevent misinterpretation, we decided to clearly mark them in grey. We would like to keep the data in the manuscript as presented in the original manuscript, also referring to the other four referees who accepted this presentation.

BGD

---

## Author Comment (AC3) · 5 Mar 2020

RC2 Comments by Referee #2, 15.01.2020

Introduction

This is an interesting analysis estimating the contribution of leaf litter on soil organic matter formation of each soil layers. Generally, this is a well performed field study on a relevant subject. The manuscript is quite interesting and decently written, although some descriptions and conclusions are inaccurate. I suggest revisions to address some of the issues I raise below.

Author response

We thank the referee for his positive feedback and the constructive comments.

1. Comment

This description is inaccurate. 0-10-cm soil sequestrated 0.99 g C m-2 yr-1 from labeled litter, 0.37 g C m-2 yr-1 in the 10-50-cm soil layers. It is not surprising, compared to the considerably large contribution of 0-10 cm soil C pools. 48% of the SOC stocks (0-180 cm) were sequestrated in the top 10 cm soil layer (Table 2).

Author response

We cannot follow this comment, as we do not see the inaccuracy of this statement. The recovered label in the MAOM fraction was calculated to average annual litter inputs (see RC5, comment #8) into the different soil compartments. Inputs were, as stated in the manuscript, highest in the topsoil, and lower in the subsoil compartments. We agree that this depth pattern is not surprising and can be expected, since it is known that highest DOC inputs occur in the mineral topsoil with a strong decline with soil depth (Leinemann et al. 2016). The same is true for inputs from a recent litter layer (Fröberg et al. 2007). We did not make changes to the statement in the original manuscript.

Leinemann, T., Mikutta, R., Kalbitz, K., Schaarschmidt, F. and Guggenberger, G.: Small scale variability of vertical water and dissolved organic matter fluxes in sandy Cambisol subsoils as revealed by segmented suction plates, Biogeochemistry, 131(1–2), 1–15, doi:10.1007/s10533-016-0259-8, 2016.

Fröberg, M., Jardine, P. M., Hanson, P. J., Swanston, C. W., Todd, D. E., Tarver, J. R. and Garten, C. T.: Low Dissolved Organic Carbon Input from Fresh Litter to Deep Mineral Soils, Soil Sci. Soc. Am. J., 71(2), 347, doi:10.2136/sssaj2006.0188, 2007

2. Comment

Lines 34-36: Most studies focused on SOC dynamics only in 0-10 cm soil layers? The

concepts of "topsoil" and "subsoil" are confusing throughout the text. According to my understanding, the authors described the soils in the 10-to-180-cm layers as "subsoil" involving their own results. But the topsoil described here is obviously not 0-10 cm only.

Author response

It is true that the soil at our study site only has a shallow topsoil horizon, which was classified as such by using the guidelines of the International WRB and the German soil classification. Topsoil horizons are defined as surface mineral soil horizons that are either enriched in organic materials or depleted in inorganic materials (i.e. by podsolization or lessivation). In the soils under study, this refers to the genetic soil horizons AE (0-to 10 cm soil depth, see Table 1). Hence, indeed the topsoil is shallow and does not exceed 10 cm. For our study site and the soil cores, it was reasonable to define the increments 0-10 cm as the topsoil increments based on the soil horizon classification. The subsoil increments (horizons B and C in our case) were divided in 3 subsections for practical reasons. The deep mineral subsoil was defined as the soil > 100 cm soil depth, since classic soil C surveys usually draw the line at 100 cm (Jobbagy and Jackson, 2000). Reversely, we considered the increments from 10 to 50 cm as the upper subsoil. Additionally, the bulk data allowed the presentation of results from the increments in between (50 to 100 cm), which were accordingly summarized as mid subsoil. This definition was given in lines 108 to 110 of the original manuscript, and reads: "Depth increments of the soil cores taken from 0-5 and 5-10 cm are defined as "topsoil", increments between 10 and 50 cm as "upper subsoil", those between 50 to 100 cm as "mid subsoil", and increments below 100 cm as "deep subsoil".

We mentioned studies on topsoil C inventories in lines 34 to 36 to introduce the reader to the topic. But of cause the topsoils in the given studies are not restricted to a soil depth of 0-10 cm, as it is a genetic criterion, dependent on the study site. Thus, it is not correct to draw the conclusion that the term "topsoil" always implies a soil depth of 0-10 cm.

Jobbagy, E. G. and Jackson, R. B.: The Vertical Distribution of Soil Organic Carbon and Its Relation to Climate and Vegetation, Ecol. Appl., 10(2), 423–436, doi:10.2307/2641104, 2000.

3. Comment

Lines 38-41: This statement is not correct. Below-ground inputs may more important contribution than the litter for SOC accumulation (Nadelhoffer and Raich, 1992; Majdi, 2001; Pausch and Kuzyakov, 2018). Nadelhoffer, K. J., and Raich, J. W.: Fine root production estimates and belowground carbon allocation in forest ecosystems. Ecology, 73, 1139–1147, 1992. Majdi, H.: Changes in fine root production and longevity in relation to water and nutrient availability in a Norway spruce stand in northern Sweden. Tree Physiol., 21, 1057–1061, 2001. Pausch, J., and Kuzyakov, Y.: Carbon input by roots into the soil: quantification of rhizodeposition from root to ecosystem scale. Glob. Change Biol., 24, 1–12, 2018.

Author response

We agree and modified the sentence from line 39 to 41 in the original manuscript as follows: "Dissolved organic matter was estimated to contribute about 19 to 50 % to the total mineral soil C stock in forest soils (Kalbitz and Kaiser, 2008, Sanderman and Amundson, 2008) and is considered as a main source of subsoil OM in temperate forest soils (Kaiser and Guggenberger, 2000), next to belowground inputs (Nadelhoffer and Raich, 1992; Majdi, 2001) ." We did not include the Pausch and Kuzyakov (2018) suggestion, because they had their main focus on crop- and grasslands and not on forest soils.

4. Comment

Lines 64-66: I noticed and agreed with the comments from Paul Hanson. And: Guelland K, Esperschütz J, Bornhauser D, et al. Mineralisation and leaching of C from 13C labelled plant litter along an initial soil chronosequence of a glacier forefield. Soil Biology and Biochemistry, 2013, 57: 237-247. Kammer A, Schmidt M W I, Hagedorn F. Decomposition pathways of 13 C-depleted leaf litter in forest soils of the Swiss Jura. Biogeochemistry, 2012, 108: 395-411.

Author response

We thank the referee for the additional references. As described in our reply to Paul Hanson's comment, we included the references in the manuscripts. In line 63 to 64 of the original manuscript. The detailed response can be found in the reply to Paul Hanson's comment.

5. Comment

Line 116: It's important to measure the mass of litter (both for initial and after 22-months) for estimating the relative contribution of the sequestrated C from litter? This is my primary concern.

Author response

We agree with the referee's opinion that the masses of the labeled litter for both time points are useful information. We already addressed this issue in RC1, comment #2 and #9.

6. Comment

Line 217: SOC content in 0-10 cm soil (8.2% here) is largely different (> 5 times) from that given in Table 1 (1.5%, the same forest plot or stand, their previous study). Is there any special on the location of the soil sampling in this study?

Author response

We agree that the discrepancy between the data in Table 1 and the soil core bulk data is confusing to the reader. To clarify this, it should be noted that the 8.2 % in our study is actually the value for the increment 0-5 cm only (visible in Fig. 1a). We recognize the misunderstanding, since we stated in line 217 " … from about $82\pm57$ mg g-1 in

the topsoil to . . .". We corrected this sentence in the following way: "Soil OC contents decreased strongly from about 82±57 mg g-1 in the upper topsoil increment (0-5 cm) to 3±1 mg g-1 in the upper subsoil at 50 cm soil depth (Fig. 1a)."

When comparing the soil core bulk data with Table 1, the mean of both increments, 0-5 and 5-10 cm should be used, which would be 5.2±3.5 % SOC compared to the 1.5 % for 10 cm thick topsoil horizon from Table 1. However, this is still a 3-fold higher SOC content for the same study site, which suggests a high spatial variability and the not very well defined border between the thin organic layer and the mineral soil.

————————————————————

---

## Author Comment (AC4) · 5 Mar 2020

RC3 Comments by Referee #1, 19.01.2020

Introduction I

This is a nice straightforward presentation of a field study investigating the fate of surficial litter-derived carbon as it enters and travels down the soil profile. The introduction presents a good overview of the current scientific understanding and of the study objectives.

Author response

[Figure]

We thank the referee for the efforts and the positive feedback. We further appreciate the comments on alternative set-ups of the experiment.

1. Comment

As mentioned by previous reviewers, it may benefit from acknowledging past studies using radioactive carbon, as well as the few studies using stable carbon to follow the fate of surface litter

Author response

Yes, we agree with this view and modified the introduction in line 63 to 64 to acknowledge previous studies. The detailed response can be found in the reply to Paul Hanson's comment.

Introduction II

The methodological approach is described in sufficient detail, and the results are concisely presented (thank you!). This paper presents a case study-results from a specific soil. There is still value in getting the work published as is, as I agree with the authors that quantitative information on the fate of carbon inputs after they enter soils is still mostly missing.

2. Comment

Out of curiosity, why was that particular study site chosen? For convenience, or was there another more scientific reason?

Author response

The study site was chosen for several reasons. One was that the Research Unit "SUB-SOM" involved 9 institutions and groups spread throughout Germany. It was relevant that the location was close to one of the central labs, the Institute of Soil Science in Hannover, where the weekly taken samples were analyzed. Another important aspect was that in a comprehensive pre-exploration of potential study sites, the Grinderwald
proved to be suitable regarding water flow conditions (e.g. high sand content, not too dense, no stagnating water, in sum good water flow conditions) and C distribution (e.g. moderate C in the mineral soil) in the soil profile. Further, we looked for a site with no land-use change during the last century and for an old-growth stand > 100 yrs. And finally, we needed to get permit from the Forestry Administration to install all the equipment and conduct the experiments.

3. Comment

However, I have been trying to wrap my head around the potential broader significance of the presented study. The studied site seems to be affected (to a large extent?) by bioturbation, and a lot of recent carbon was recovered in particulate organic matter. How would the situation be different in the case of soils less affected by soil fauna? Not only in term of the topsoil carbon, but also more importantly in term of DOC leaching and redistribution lower in the profile? Would fluxes then be more important?

Author response

The bioturbation was largely restricted to the top 0-10 cm. We assume that less mixing of POM into the mineral soil would result in an initially higher sequestration of C in the organic layer, e.g. due to retention by the organic layer itself as it was shown by Fröberg et al. (2007) for a coniferous forest floor. It can be expected that if this material would have stayed on the mineral soil, is likely faster decomposed to $CO_2$. Concerning this effect on DOC formation and leaching we can only speculate. But in absolute means, the amount of litter translocated to the mineral soil by DOC is small (about 2 % of the applied litter after 22 months).So the effect on DOC formation and leaching should be also very minor.

Fröberg, M., Berggren Kleja, D. and Hagedorn, F.: The contribution of fresh litter to dissolved organic carbon leached from a coniferous forest floor, Eur. J. Soil Sci., 58(1), 108–114, doi:10.1111/j.1365 2389.2006.00812.x, 2007

**4. Comment**

Lastly, how would the results look like if the study had been conducted longer? Eighteen months may not be enough time to see redistribution at depth.

Author response

We agree that 18 months likely is not sufficient to detect a considerable translocation as a result of the assumed sorption-microbial processing-desorption cycles form the litter layer down to the deep subsoil. If we think of prolonging the experiment with the exact setting as we used it, i.e. level of 13C enrichment in the labeled litter, we assume that the continuous input of new and unlabeled compounds will rather soon shift the measurable enrichments towards the natural abundance, as we already saw it in the second sampling of our experiment.

---

## Author Comment (AC5) · 5 Mar 2020

RC4 Comments by Referee #4, 20.01.2020

Introduction

This manuscript is well written and organised. At this stage I have comments/concerns regarding some methodological aspects which are elaborated below.

Author response

We want to thank the referee for the positive feedback and the constructive criticism, which helped to improve the manuscript.

**1. Comment**

Also, the points touched on in the discussion are clear, but don't bring the arguments back (explicitly) to the hypotheses presented in the Introduction.

Author response

We appreciate the referees comment on the discussion of the hypotheses. We agree that a direct answer to the hypothesis of the original version of the manuscript was not given. We modified the manuscript by changing hypotheses to questions from lines 70-80 of the original manuscript as follows:

"Particularly, we aim at answering the following questions:

1. Does recent aboveground litter significantly contribute to the accumulation of OM in subsoils?

2. Is OM transferred into the subsoil directly via the DOM pathway, or is subsoil OM the result of repeated sorption-microbial processing-desorption cycles?

3. To which extent is recent aboveground litter-derived C sorbed to soil minerals and does this fraction represent a source of stable SOM?

To quantify the contribution of recent litter to subsoil C stocks via DOM movement and evaluate the stability of litter-derived SOM, we... "

With the following comparison, we want to point out that we provided an answer to all three questions in our implications:

Lines 384-386: "In fact, we did not find a translocation of considerable amounts of recent litter-derived C into the deep subsoil, indicating that most translocated OM at the study site is of older age." - This implication answers question 1.

Lines 386-387: "Our field study supports the concept that C accumulation in deeper soil involves several (re)mobilization cycles of OM during its downward migration." -

This implication answers question 2.

Lines 389-390: "Slowest turnover of litter-derived C was observed for MAOM compared to both POM fractions, supporting the assumption that accessibility and sorptive stabilization reduces the vulnerability of OM to microbial decomposition." - This implication answers question 3.

2. Comment

How was the highly labelled litter (i.e. the source of the enriched C), produced? Is it homogeneously labelled? This is important because if labelling is not homogeneous only some compound types and pools of C will be traceable, which may not represent the whole plant C well, or bias it against the movement and stabilization of certain litter-derived compounds. This could lead to substantial underestimation/over estimation (?) of the contributions or surface litter. It would also affect the overall estimation of loss. Given the type of goal, which is mainly one of quantifying contribution (vs. comparing different treatments) this is of high importance and is potentially concerning.

Author response

Thanks for this important question. The highly labeled litter (10-14 at% 13C) was purchased at IsoLife, a company which is specialized on labeling plants by growing them in greenhouses under a 13CO2-enriched atmosphere. The labeling in the 13CO2 atmosphere was long-term and continuous, thus It can be expected that the label is homogeneously distributed in all plant compartments. We added this information to line 100 of the original manuscript as follows: "Labeled litter was prepared as a mixture of highly labeled beech litter (10 atom-% uniformly labeled due to growth under 13CO2-enriched atmosphere in a greenhouse, IsoLife, Wageningen, The Netherlands)..."

3. Comment

Question: how were the labelled and unlabelled litter mixed?

Author response

**BGD**

The litter types were mixed at a certain ratio as intact leaves (dried). By keeping them intact, we accepted that a 100 % homogeneous distribution on the plot at small scale was unlikely, but we wanted that the litter application resembled a fresh litterfall. To account for the potential heterogeneity on the cm scale, three cores were drilled per plot and composite samples were prepared and used for analysis and fractionation (as presented in lines 113 to 114).

4.1 Comment

I am assuming that the natural litter added is "fresh" litter? Or at what stage of decay is?

Author response

Yes, we added dried undecomposed litter.

4.2 Comment

And what about the labelled litter? Is it senesced? Fresh? If the two litters were are different stages, this would have implications, because of the differential composition of C pools, depending on the potential scenarios. In this case, what could they be? This is also a consideration for the initial mixed litter.

Author response

We agree that different decomposition stages would massively influence the intended homogenous leaching of DOM from both litter components (labeled and natural). We considered that and have chosen litter in the same stage, meaning leaves after senescence but before shedding for both types of litter, and from very young trees in the field.

5. Comment

I don't understand how the 20mm (do you mean 2 cm mesh?) could prevent the leaching of the naturally fallen litter to reach the soil

Author response

The mesh, with its mesh size of 2 cm, was not installed to prevent leaching (this would imply a complete water blockage of the area, destroying the natural conditions). The mesh had two main functions. First, it prevented translocation of the labeled litter by wind, potentially onto the control sites. Second, it allowed us the removal of freshly fallen leave litter in autumn after the experiment started, in order to avoid a dilution of the 13C signal in the following year. Since the former explanation was not given in the submitted version of the manuscript, we now added this to line 101 of the original manuscript as follows: "A net (2 cm mesh size) was installed on top of the litter layer to, first, prevent surface translocation by wind, and second, to avoid dilution of the labeled litter over time by the seasonally fallen litter."

6. Comment

I am confused by the handling of the samples for water extractions. Line 115 says they were soil subsamples were frozen directly after sampling for water extractions, but later one it says field-fresh samples were extracted. The freezing and thawing will have an impact on the C composition of the soil solution from the breaking of the microbial cells, putting cellular contents into solution, potentially. Then also, if the soils were not extracted soon after field collection the C composition of the soil solution and its isotopic composition would potentially change too. With such the low levels of enrichment that reach the sub-soil, these unintended impacts of the handling could alter the results.

Author response

We agree with the reviewer, that the term "field fresh" is confusing and not correct. In fact, the samples were kept frozen and after storage thawed for 24 h at 4°C. Thereafter the samples were sieved (< 2 mm) and then extracted with 1 mM CaCl2 solution. We see the point that freezing and thawing might have an impact on the C composition of the water extracts. However, we decided to freeze the samples to treat them equally. The assumption behind this was, if all samples were stored in the fridge at

4°C, microbial turnover would be still active. Furthermore, due to the large amounts of samples (n=90) we were not able to extract them all after the same time of storage. In consequence, there would also be a bias due to the different storing time in the fridge. Therefore, we decided to freeze all samples. We added this information to the original manuscript at line 129 as follows: "Prior to the extraction, the frozen samples were thawed for 24 hours at 4°C and thereafter sieved to < 2 mm. Following the procedure of Chantigny et al. (2006), [...]."

7. Comment

It would be good to explain the general purpose of the investigation of HF surfaces in the methods.

Author response

We agree that a short description of the general purpose of the HF surface investigations is helpful for the reader to make use of the data in the supplement. We added the following sentence to line 175 of the original manuscript: "Surfaces of the HF were further investigated by X-ray photoelectron spectroscopy (XPS) with respect to the elemental composition as a function of soil depth. Method description and data are presented in the Supplement."

8. Comment

Not methodological: In the Introduction, some potential reasons for the 13C enrichment with depth are mentioned; there are some new developments about this gradient such as evidence also suggesting there is a contribution also of the microbial composition of the necromass (e.g. Biogeochemistry 2015, 124: 13-26)

Author response

We thank the referee for this additional view, which we did not included so far in our manuscript. The aspect of a compositional change within the microbial community and its necromass may not influence the implications from our study, but we agree

that it should be mentioned in the introduction to this topic. We therefore modified the sentence in lines 51 to 53 of the original manuscript as follows: "In most soils, $\delta$13C values increase with soil depth, which is related to the isotopic discrimination of the heavier C isotopes during microbial respiration (Nadelhoffer and Fry, 1988, Balesdent et al. 1993, van Dam et al. 1997) or a shift in the fungal to bacterial ratio in favor of the more 13C-enriched bacteria (Kohl et al. 2015)."

―――――――――――――――――――――

---

## Author Comment (AC6) · 5 Mar 2020

RC5 Comments by Referee #5, 22.01.2020

Introduction

The article deals with a significant question that is the redistribution of fresh litter carbon into different soil C pools and the processes involved in the transfer and turnover of carbon in the whole soil profile. The use of stable isotope labeling in a field study seems really adapted. However some information is missing, sometimes with significant importance for interpretation.

[Figure]

Author response

We thank the referee for the efforts and the constructive criticism which helped to improve the manuscript.

1. Comment

Main concerns: What is the percentage of remaining litter (in mass and labelling) after 22 months? This data is important to assess the percentage of litter lost by mineralization compared to the part that did not enter the soil, and to know if the incorporation of litter fits well with natural conditions (11% seems low).

Author response

After 22 months, removed litter layer amounted about 405 g m-2 per plot with a remaining enrichment of 384.4 ‰ 13C. About 25 % of this removed litter originates from the initial applied labeled litter, while 75 % originates from litterfall (e.g. shells of beechnuts), which passed through the 2 mm mesh and accumulated during the 22 months. Comparing initial and removed labeled litter, we recovered about 35-40 % as residual labeled litter, while we recovered an evolution of labeled litter-derived CO2 of about 36-40 %. Adding the 11 % found in SOC, we know the whereabouts of roughly 85 % of the initial applied litter. The certain offset in this calculation represents the amount of label, which we were not able to recover. We consider these results as quite decent and well fitting for a labeling experiment under field conditions with a duration of nearly two years.

2. Comment

What are the properties of labeled and unlabeled litter?

Author response

We thank the referee for this comment. In fact, referee #3 (RC1, comment #2) and referee #4 (RC4, comment #2) commentated this as well, a detailed explanation can

be found in the respective response. The labeled and unlabeled litter were the same, except of the 13C enrichment. We did not go into detail about the properties since both types were comparable and no differences between the treatments can be expected. Distribution of the 13C in the labeled litter was homogeneous.

3. Comment

Was the labeling realized through continuous or pulse technique? In relation to this question, what is the _13C of the remaining litter after 22 months (is it consistent with the _13C measured for initial litter or may have the incorporation process been discriminant?). The interpretation of the labeling calculation could be different if the labeling is not homogenous.

Author response

The production of the highly enriched labeled litter (at IsoLife) was realized through a continuous labeling in a 13CO2 atmosphere and IsoLife assures homogeneous labeling of the leaf litter. The $\delta$13C ratio of the remaining litter was about 384 ‰ (mean of 5 replicate measurements at two different institutes). But related to the authors response to comment 1, there was dilution due to accumulation of unlabeled litter during the 22 months, thus we cannot determine if there was a discrimination during litter decomposition and leaching or not.

4. Comment

All the data that would allow comparing the properties of the different plots, especially labeled/unlabeled plots. For example, authors always averaged the control and the labeled plots for soil properties: the C contents and stocks (figure 1), the distribution in density fractions (in figures 2 and 6, the variability is high as mentioned in the caption: is it distributed randomly between labeled and unlabeled plots or is there significant differences?), the C/N (figure 3), WEOC (figure 4). If significant differences exist between the labeled/unlabeled plots, the interpretation of the low difference in isotopic signature

(figure 5) could be limited.

Author response

We tested for significance by using a t-Test. In total, we tested 149 sample subsets for MAOM C/N ratio and C content, WEOC data (WEOC in %, SUVA, HIX) and C distribution in the individual density fractions. All tests were done for each depth increment individually. We found that a total of 10 out of 149 tests resulted in significant differences, which were distributed randomly between several parameters and fractions and depth increments. Considering all tests, we recognize the potential differences between the labeled and unlabeled subplots as insignificant without further implications for our interpretations.

5. Comment

Table S1: add the value of the reference (C, N or _13C). Moreover, was a labeled standard (in-house) used since the initial enrichment applied was high (1241-1880 prmil)? What is the maximum _13C measured in the soil fractions and used for calculation?

Author response

We added all relevant values to Table S1 which were used for correction and calculation of the data. This includes the C and N data for the HOS standard and the $\delta$13C ratio for the IAEA standards. To avoid further misunderstandings, we deleted the standard substances for 15N from Table S2, since these data are not included in the manuscript. There was no labeled standard included in the measurements. This was also not considered necessary, as the $\delta$13C values of the soil fractions had a maximum of -6 ‰

6. Comment

Minor concerns: A 2-mm mesh was used to prevent new litter input during first 22 months: what about a potential leaching of additional unlabeled litter? As mentioned lines 366-368: WEOC release is possible.

Author response

The 20-mm mesh was used to reduce mixing of the labeled litter with fresh litterfall in autumn and to be able to remove that fresh litter in late autumn to prevent a massive interference of unlabeled litter in the following 12 months of the experiment. Hence, there was just limited leaching from the material lying on the mesh until it was removed in a weekly interval. But as was replied to a similar question raised in RC1, comment #9, litter <20 mm (e.g. fruits) could have passed the mesh. This also led to an amount of removed litter (after 22 months) which was larger than the initial application.

7. Comment

Was the WEOC extracted on frozen samples (line 115) or on field-moist samples (line 129)?

Author response

Water extractable OC was extracted on field-moist samples after thawing, as samples were frozen directly after sampling for storage and comparability reasons. We responded in detail to a similar question in RC4, comment #6. We added the missing information about the thawing process to line 129 of the original manuscript. It reads: "Prior to the extraction, the frozen samples were thawed for 24 hours at 4°C and thereafter sieved to < 2 mm."

8. Comment

Line 322-325: the sentence is not clear for me. Which recalculation was done? Is it to correct the input of litter to the soil of the experiment that was not representative of typical "annual" litter input? If it is the case: what about this difference (line 98-99, authors mentioned a "equivalent amount of litter" added to the plots)?

Author response

In lines 322 to 325 we wanted to express that the amount of labeled litter-derived C in

the MAOM fraction of different depths after 22 months was used to estimate an incorporation rate per year. The data basis is the litter-derived C in g m-2 at the first sampling, as it is given in Fig. 6a. The data of each individual increment were cumulated with respect to our 3 main soil compartments topsoil, upper subsoil and deeper subsoil. Data for the 3 compartments were then recalculated to a yearly basis (12 months/22 months). We are aware of the implications of this recalculation, which is that a linear incorporation of litter-derived C over the 22 months is assumed. This is likely not the case, as the initial translocation and incorporation (first weeks) may be higher than after 20 or 22 months, but in the end we are stating an estimate. We added this assumption to line 325 of the original manuscript as follows: "For the Dystric Cambisol under European beech, the observed recoveries of 13C in MAOM in the 22 months of labeled litter application were recalculated to average annual litter inputs from the recent litter layer into the HF of about were estimated as 0.99 $\pm$ 0.45 g C m-2 yr-1 in the topsoil, 0.37 $\pm$ 0.10 g C m-2 yr-1 in the upper subsoil, and 0.01 $\pm$ 0.01 g C m-2 yr-1 in the deeper subsoil. This estimation follows the assumption of a constant input of labeled litter-derived OM during the 22 months, which is a sufficient approximation for this estimate but may not reflect the actual conditions in the field.

9. Comment

The XPS part does not seem to be related to the study. Is it necessary? Additional Table S3, Figure S1, figure S3 and figure S6 are not cited in the text for example. At least Table S3 and Figure S5 seem redundant. I would delete the Table S3 (or simplify it? The replicates should be averaged).

Author response

We appreciate the referee's notice about relevance and citations of the supplementary material. We agree that a presentation of the detailed XPS results in the form of a table (Table S3) may not be necessary, since the most relevant part of the data is also given as a more concisely figure (Fig. S5). But we see a benefit in showing these

data, because sorption of C to minerals and the formation of MAOM is taking place on mineral surfaces. Consequently, investigations of the surface composition of the HF is a helpful tool. The data given in Fig. S5 for example show and also validate that surface C and N contents are decreasing with soil depth as seen in the EA-IRMS analysis. Vice versa, higher contents of Fe and Al were found with increasing soil depth, resembling a higher proportion of uncovered mineral surfaces and, in theory, potentially available sorption sites. Due to their value as additional information, we suggest to keep the XPS analysis included. However, we agree to reduce its presence in the supplement by deleting Table S3 and Fig. S6. We added a citation of Fig. S1 to line 226 of the original manuscript: "...5 in the deep subsoil (Fig. 3b), similarly to the bulk soil C/N (Fig. S1)." We added a citation of Fig. S3 to line 241 of the original manuscript: "..., whereby the contribution of the deeper depth increments was very minor (Fig. 6a, Fig. S3)."

10. Comment

Line 131: Were the filters pre-rinsed? Was the effect of cellulose pollution on _13C of WEOC assessed? Is it negligible (depending on the DOC content of the WEOC extract)? If this pollution is equivalent for the labeled/unlabeled WEOC (and if WEOC is equivalent for labeled/unlabeled plots), it may not impact the isotopic calculation but should be mentioned.

Author response

Yes the filters were pre-rinsed with 250 ml of the 1mM CaCl2 solution. In a pre-test we tested for the pollution by the filters, but the effect was negligible. However, for each extraction we also run 2 blank samples (only the CaCl2 solution) with all steps (shaking, centrifuging, extraction). The TOC of the blanks were used for the correction of TOC from the water extracts. And we also determined the isotope ratio of the blanks, but they had a similar signature as WEOC (control). We added the missing information to line 132 of the original manuscript as follows: "Prior to the filtration, filters were

pre-rinsed with 250 mL of the 1 mM CaCl2 solution".

11. Comment

The WEOC extraction and the density fractionation were done in parallel (not sequential). So is there a relation between the DOC collected in SPT fraction and the WEOC? Line 227, authors mentioned that the "no consistent trend" was observed for DOC of SPT. What about the _13C of SPT?

Author response

In previous studies, e.g., Gentsch et al. (2018) we could show that SPT solution is one relevant pool of C, which should be considered and measured after density fractionation. Especially if the aim is to set up a carbon balance, neglecting the carbon which dissolves in the SPT solution during fractionation will automatically lead to a loss of several % C, one of the reasons why we list the SPT fraction next to the soil fractions. From a methodological point of view, we consider a comparison and relation to WEOC as possible but not useful, since the functionality behind the WEOC fraction (e.g. potential to be mobilized and translocated by the soil solution, bioavailability) cannot be transferred to the CSPT fraction. The main reason is that the soil is under extreme conditions during fractionation (high salt contents, ultrasonic dispersion, and a higher soil to water ratio), likely mobilizing C, which would not be mobilizable under natural conditions. With regards to our results, we also do not see a close relation, as WEOC proportions consistently increased with soil depth (Fig. 4a), while proportions of CSPT (DOC in SPT) varied between 1 to 3 % but without a depth trend (Fig. 2). $\delta$13C values in the CSPT fraction were lower than those in the WEOC, the same is true for the recovered labeled litter-derived C. This may likely imply that the density fractionation treatment mobilized a higher amount of older and better stabilized C compared to the water extraction.

Gentsch, N., Wild, B., Mikutta, R., Čapek, P., Diáková, K., Schrumpf, M., Turner, S., Minnich, C., Schaarschmidt, F., Shibistova, O., Schnecker, J., Urich, T., Gittel, A.,

Šantrůčková, H., Bárta, J., Lashchinskiy, N., Fuß, R., Richter, A. and Guggenberger, G.: Temperature response of permafrost soil carbon is attenuated by mineral protection, Glob. Change Biol., 24(8), 3401–3415, doi:10.1111/gcb.14316, 2018.

12. Comment

Line 169-174: Measurement of nitrate and ammonium, for calculation of organic nitrogen were mentioned, but never used in result. Delete?

Author response

Nitrate and ammonium data were used for correcting the N-contents of the bulk data and MAOM before calculating the C/N ratios (Fig. 3b, Fig. S1). But we see the potential misunderstanding and modified the respective figure captions by adding this information to line 641 of the original manuscript and line 113 of the supplementary material. It now reads as "Nitrogen contents in the HF were corrected for extractable nitrate and ammonium contents.".

13. Comment

Line 174-175: Is the XPS really useful for this study?

Author response

This question is related to Referee comment 9 and the reader is referred to the authors' response given there.

14. Comment

Line 228: cite Figure 2

Author response

We added a citation of Fig. 2.

15. Comment

Line 248: cite Figure 5c

Author response

We added a citation of Fig. 5c.

16. Comment

Line 258: cite Figure 6b

Author response

We added a citation of Fig. 6b.

17. Comment

Line 267: Sentence is not clear: what means "similar loss for recovered material", is it 77% of mass or of carbon?

Author response

The 77 % represent the loss of labeled litter-derived C in the bulk soil samples when comparing the first with the second sampling, similar to our statement of losses in the different functional fractions of 66 to 89 % in the sentence before. We were mentioning the bulk soil losses in this context in order to give the reader an impression about the decent consistency of soil fractions and bulk data. For clarification, we modified the statement in line 266 to 267 of the original manuscript as follows: "The decline of label from mass-weighted individual OM fractions was similar in magnitude to the loss of labeled litter-derived C in the bulk samples (77 %; data not shown)."

18. Comment

Table 3: add the % of initial litter that was lost by mineralization compared to the material remaining after 22 months. Express the values in % of C "entering" the soil.

Author response

We added the information of the initial litter and its loss due to CO2 respiration in the Table 3 caption (it no reads as: "Overall, 36-40 % of the initially applied litter was lost by respiration during 22 months of field exposure (Wordell-Dietrich, unpublished).") as well as in the text in lines 102 to 103 in % of the initial label (it now reads as: "A total of about 36-40 % of the initially applied labeled litter-C left as CO2 (Wordell-Dietrich, unpublished)."). By comparison with the amounts of recovered labeled litter-derived C in the soil profile, it will be evident for the reader that CO2 respiration is more than 3-times as high as incorporation in the soil.

19. Comment

Figure 1, 3, 4: show the mean and SD of labeled and unlabeled plot, of 22 and 40 months.

Author response

We checked labeled and unlabeled plots for significant differences (by use of a t-Test) as it was mentioned in the Authors response to comment no. 4. Since differences were insignificant for the vast majority of tests and the only difference between labeled and unlabeled plots (in treatment and sample processing) was the label application, we rather consider labeled and unlabeled samples as field replicates for all non-isotopic parameters.

---

## Author Response (AR1)

**Interactive comment(s) on "Relevance of aboveground litter for soil organic matter formation – a soil profile perspective" by Patrick Liebmann et al.**

We want to thank all referees and appreciate the comments from the scientific community. In the following, the response to all comments is given in the order of appearance. The proposed changes to the manuscript due to the respective referee comment are marked in red.

| SC1                | Short comment by Paul Hanson, 08.01.2020                                                                                                                                                                                                                                                                                                                                                                                                                                                                                                                                                                                                                                                                     |
|--------------------|--------------------------------------------------------------------------------------------------------------------------------------------------------------------------------------------------------------------------------------------------------------------------------------------------------------------------------------------------------------------------------------------------------------------------------------------------------------------------------------------------------------------------------------------------------------------------------------------------------------------------------------------------------------------------------------------------------------|
| 1. Comment         | The following 5 papers report on field studies of enriched background 14C isotopic tracers for multiyear controlled litter additions and the transfer of those labels into the soil. You might modify your statement on line 65 to recognize these efforts. The Kramer et al. 2010 paper is probably the most relevant, and you have already cited the related mesocosm study (Froberg et al. 2009) 1. Tipping E, Chamberlain PM, Fröberg M, Hanson PJ, Jardine PM (2012) Simulation of carbon cycling, including dissolved organic carbon transport, in forest soil locally enriched with 14C. Biogeochemistry 108:91-107, doi 10.1007/s10533-011-9575-1.                                                   |
|                    | 2. Parton WJ, Hanson PJ, Swanston C, Torn M, Trumbore SE, Riley W, Kelly R (2010) ForCent model development and testing using the Enriched Background Isotope Study (EBIS) experiment. JGR-Biogeosciences 115:G04001, doi:10.1029/2009JG001193                                                                                                                                                                                                                                                                                                                                                                                                                                                               |
|                    | 3. Kramer C, Trumbore S, Fröberg M, Cisneros-Dozal LM, Zhang D, Xu X,
Santos G, Hanson PJ (2010) Recent (<4 year old) leaf litter is not a major source
of microbial carbon in a temperate forest mineral soil. Soil Biology and
Biochemistry 42:1028-1037.                                                                                                                                                                                                                                                                                                                                                                                                                                         |
|                    | 4. Riley WJ, Gaudinski JB, Torn MS, Joslin JD, Hanson PJ (2009) Fine-root mortality rates in a temperate forest: estimates using radiocarbon data and numerical modeling. New Phytologist 184:387-398.                                                                                                                                                                                                                                                                                                                                                                                                                                                                                                       |
|                    | 5. Fröberg M, Hanson PJ, Trumbore SE, Swanston CW, Todd DE (2009) Flux of carbon from 14C-enriched leaf litter throughout a forest soil mesocosm. Geoderma 149:181-188. [Mesocosm study in support of the larger field EBIS effort.]                                                                                                                                                                                                                                                                                                                                                                                                                                                                         |
| Author
response | We agree and appreciate the suggested references of Paul Hanson and we will add
some of the literature he suggested here. We received additional suggestions from
Referee #2 (RC2, comment 4) and decided to include just a selection of Paul
Hansons and Referee #2s suggestions, in order to satisfy both comments and to
limit to a maximum of three references per citation. We just want to note that our
main focus (and also novelty of the study) is in the subsoil aspect. The suggested
publications all have their relevance for discovering the fate of litter layer-C and
we will include them here, but they mostly cover the topsoil or a soil depth of 0-10
cm only. |
|                    | We modified the sentence in lines 63 to 64 of the original manuscript as follows:
"In order to quantify individual C fractions and fluxes, isotope labeling, e.g. using
13 C- or 14 C-enriched litter material, has been proven as a very powerful tool (Bird
et al., 2008, Moore-Kucera and Dick, 2008, Kramer et al. 2010). Extensive
retention of DOC in topsoil horizons has been documented for field-exposed
mesocosms (Fröberg et al. 2009) or in field approaches (Kammer et al. 2012).                                                                                                                                                                         |

| RC1          | Comments by Referee #3, 10.01.2020                                                                         |
|--------------|------------------------------------------------------------------------------------------------------------|
| Introduction | This study investigated the impact of aboveground litter for soil organic carbon                           |
|              | (C) sequestration and the subsequent partitioning of litter-derived C in different                         |
|              | soil layers and OM fractions. In general, I think the data are solid and the results                       |
|              | are valuable for understanding fates of litter C input. I have some minor                                  |
|              | comments/suggestions that could improve the manuscript.                                                    |
| Authors      | We want to thank the referee for his positive feedback and the helpful and                                 |
| response     | constructive criticism, which helped to improve the manuscript.                                            |
| 1. Comment   | Lines 37 and 38:                                                                                           |
|              | This statement may be correct only for natural ecosystems. For example, OM                                 |
|              | disturbance due to tillage may be a pathway for cropping systems.                                          |
| Author       | We agree with the Referees comment and modified the sentence in line 37 to 38 of                           |
| response     | the original manuscript as follows:                                                                        |
|              | "In forest ecosystems, major pathways of OM to enter subsoils are                                          |
|              | rhizodepositions, root exudation and dissolved organic matter (DOM) leached                                |
|              | rom the horizons above (whithson et al., 2009; Rumpel and Kogel-Knabher, 2011; Vaigar and Kalbitz, 2012)." |
| 2 Commont    | Line 00:                                                                                                   |
| 2. Comment   | Lille 77.
Did you also observe the amount and chemical properties of the litter? These                  |
|              | factors could impact litter decomposition and are important for interpreting the                           |
|              | results                                                                                                    |
| Author       | The amount of litter was about $275 \text{ g m}^{-2}$ This was defined according to                        |
| response     | measurements of Meier et al. (2005) who reported a litter input of a beech forest                          |
| response     | at 165 of 427 g m -2 . Chemical properties were not analyzed, but since the labeled             |
|              | leaves were harvested from the same tree type ( Fagus sylvatica ) as in the research                |
|              | forest, we assume that the chemical properties of the litter resemble the natural                          |
|              | environment of our study site.                                                                             |
|              | We recognize this comment and modified the sentence in line 98 to 99 of the                                |
|              | original manuscript as follows:                                                                            |
|              | "For the labeling, the natural litter layer was removed manually and replaced by                           |
|              | an equivalent amount of 275 g 13 C enriched beech litter per m -2 , representing a   |
|              | typical input of beech litter in Germany (Meier et al., 2005). Labeled litter was                          |
|              | prepared as"                                                                                               |
| 3. Comment   | Line 271: "both, inputs".                                                                                  |
| Author       | We are unsure about the intention of this comment, but we assume it aimed at the                           |
| response     | comma? But manuscript text and comment are the same. No changes to the                                     |
|              | sentence in the original manuscript were made.                                                             |
| 4. Comment   | Line 301:                                                                                                  |
| A .1         | For "DOM", did you mean DOM leached from surface soil layers?                                              |
| Author       | In the first passage of this sub-chapter 4.2 (from line 290 to 314), we discuss the                        |
| response     | overall role of DOM for MAOM formation, without a specific focus on litter-                                |
|              | derived DOM but rainer relate the DOM in general, i.e. of different source. To                             |
|              | prevent possible misunderstandings, the sentence in the original manuscript was                            |
|              | "Decomposition of roots can substantially contribute to the subsoil SOM pool as                            |
|              | well (Rasse et al. 2005), but since root density (Heinze et al. 2018; Wordell-                             |
|              | Dietrich et al. 2019) and root evudation (Tückmantel et al. 2017) are low in the                           |
|              | Grinderwald subsoil, we assume that the increasing share of MAOM with soil                                 |
|              | depth rather suggests an increasing importance of DOM as a dominant source of C                            |
|              | in this forest subsoil, irrespective of its origin."                                                       |
| 5. Comment   | Is it possible that rhizodeposition still made a considerable contribution in subsoil                      |
|              | MAOM although root density and exudation were low. given that subsoil MAOM                                 |
|              | contents were also very low?                                                                               |
| Author       | A considerable contribution of rhizodepositions is possible, as we also found that                         |
| response     | about 20 % of the deep subsoil SOC is present as POM (Fig. 2), most likely                                 |

|            | derived from roots, but we assume that it is not the dominant source of subsoil                                                                                                                                                                                                                                                                                                                                                                                                                                                                                                                                                                                                                                                                                                                                                                                                                                                                                                                                                                                                                                                                                                                                                                                                                                                                                                                                                                                                                                                                                                                                                                                                                                                                                                                                                                                                                                                                                                                                                                                                                                                                                                                                                                                                                                                                                                                                                                                                                                                                                                                                                                                                                                                                                                                                                                                                                                                                                                                                                                                                                                                                                                                                                                                                                                                                                                                                                                                                                                                                                                                                                                                                                                                                                                                                                                                                                                                                                                                                                                                                                                                                                                                                                                                                                                                                                                                                                                                                                                                                                                                                                                                                                                                                                                                                                                                                                                                                                                                                                                                                                                                         |
|------------|-----------------------------------------------------------------------------------------------------------------------------------------------------------------------------------------------------------------------------------------------------------------------------------------------------------------------------------------------------------------------------------------------------------------------------------------------------------------------------------------------------------------------------------------------------------------------------------------------------------------------------------------------------------------------------------------------------------------------------------------------------------------------------------------------------------------------------------------------------------------------------------------------------------------------------------------------------------------------------------------------------------------------------------------------------------------------------------------------------------------------------------------------------------------------------------------------------------------------------------------------------------------------------------------------------------------------------------------------------------------------------------------------------------------------------------------------------------------------------------------------------------------------------------------------------------------------------------------------------------------------------------------------------------------------------------------------------------------------------------------------------------------------------------------------------------------------------------------------------------------------------------------------------------------------------------------------------------------------------------------------------------------------------------------------------------------------------------------------------------------------------------------------------------------------------------------------------------------------------------------------------------------------------------------------------------------------------------------------------------------------------------------------------------------------------------------------------------------------------------------------------------------------------------------------------------------------------------------------------------------------------------------------------------------------------------------------------------------------------------------------------------------------------------------------------------------------------------------------------------------------------------------------------------------------------------------------------------------------------------------------------------------------------------------------------------------------------------------------------------------------------------------------------------------------------------------------------------------------------------------------------------------------------------------------------------------------------------------------------------------------------------------------------------------------------------------------------------------------------------------------------------------------------------------------------------------------------------------------------------------------------------------------------------------------------------------------------------------------------------------------------------------------------------------------------------------------------------------------------------------------------------------------------------------------------------------------------------------------------------------------------------------------------------------------------------------------------------------------------------------------------------------------------------------------------------------------------------------------------------------------------------------------------------------------------------------------------------------------------------------------------------------------------------------------------------------------------------------------------------------------------------------------------------------------------------------------------------------------------------------------------------------------------------------------------------------------------------------------------------------------------------------------------------------------------------------------------------------------------------------------------------------------------------------------------------------------------------------------------------------------------------------------------------------------------------------------------------------------------------------------------------------|
|            | MAOM, as we discussed this in lines 298 to 301 of the original manuscript. And,                                                                                                                                                                                                                                                                                                                                                                                                                                                                                                                                                                                                                                                                                                                                                                                                                                                                                                                                                                                                                                                                                                                                                                                                                                                                                                                                                                                                                                                                                                                                                                                                                                                                                                                                                                                                                                                                                                                                                                                                                                                                                                                                                                                                                                                                                                                                                                                                                                                                                                                                                                                                                                                                                                                                                                                                                                                                                                                                                                                                                                                                                                                                                                                                                                                                                                                                                                                                                                                                                                                                                                                                                                                                                                                                                                                                                                                                                                                                                                                                                                                                                                                                                                                                                                                                                                                                                                                                                                                                                                                                                                                                                                                                                                                                                                                                                                                                                                                                                                                                                                                         |
|            | yes, in absolute number, the content of subsoil MAOM is very low (Fig. 3a). In                                                                                                                                                                                                                                                                                                                                                                                                                                                                                                                                                                                                                                                                                                                                                                                                                                                                                                                                                                                                                                                                                                                                                                                                                                                                                                                                                                                                                                                                                                                                                                                                                                                                                                                                                                                                                                                                                                                                                                                                                                                                                                                                                                                                                                                                                                                                                                                                                                                                                                                                                                                                                                                                                                                                                                                                                                                                                                                                                                                                                                                                                                                                                                                                                                                                                                                                                                                                                                                                                                                                                                                                                                                                                                                                                                                                                                                                                                                                                                                                                                                                                                                                                                                                                                                                                                                                                                                                                                                                                                                                                                                                                                                                                                                                                                                                                                                                                                                                                                                                                                                          |
|            | this paragraph, we wanted to highlight the shift in the importance of the different                                                                                                                                                                                                                                                                                                                                                                                                                                                                                                                                                                                                                                                                                                                                                                                                                                                                                                                                                                                                                                                                                                                                                                                                                                                                                                                                                                                                                                                                                                                                                                                                                                                                                                                                                                                                                                                                                                                                                                                                                                                                                                                                                                                                                                                                                                                                                                                                                                                                                                                                                                                                                                                                                                                                                                                                                                                                                                                                                                                                                                                                                                                                                                                                                                                                                                                                                                                                                                                                                                                                                                                                                                                                                                                                                                                                                                                                                                                                                                                                                                                                                                                                                                                                                                                                                                                                                                                                                                                                                                                                                                                                                                                                                                                                                                                                                                                                                                                                                                                                                                                     |
|            | functional OM fractions from 25 % to 77 % with increasing soil depth.                                                                                                                                                                                                                                                                                                                                                                                                                                                                                                                                                                                                                                                                                                                                                                                                                                                                                                                                                                                                                                                                                                                                                                                                                                                                                                                                                                                                                                                                                                                                                                                                                                                                                                                                                                                                                                                                                                                                                                                                                                                                                                                                                                                                                                                                                                                                                                                                                                                                                                                                                                                                                                                                                                                                                                                                                                                                                                                                                                                                                                                                                                                                                                                                                                                                                                                                                                                                                                                                                                                                                                                                                                                                                                                                                                                                                                                                                                                                                                                                                                                                                                                                                                                                                                                                                                                                                                                                                                                                                                                                                                                                                                                                                                                                                                                                                                                                                                                                                                                                                                                                   |
| 6. Comment | In addition, it looks that microbial decomposition of root derived C may also                                                                                                                                                                                                                                                                                                                                                                                                                                                                                                                                                                                                                                                                                                                                                                                                                                                                                                                                                                                                                                                                                                                                                                                                                                                                                                                                                                                                                                                                                                                                                                                                                                                                                                                                                                                                                                                                                                                                                                                                                                                                                                                                                                                                                                                                                                                                                                                                                                                                                                                                                                                                                                                                                                                                                                                                                                                                                                                                                                                                                                                                                                                                                                                                                                                                                                                                                                                                                                                                                                                                                                                                                                                                                                                                                                                                                                                                                                                                                                                                                                                                                                                                                                                                                                                                                                                                                                                                                                                                                                                                                                                                                                                                                                                                                                                                                                                                                                                                                                                                                                                           |
| o. comment | increases $^{13}$ C values and decrease C/N ratios of MAOM: so I am wondering if the                                                                                                                                                                                                                                                                                                                                                                                                                                                                                                                                                                                                                                                                                                                                                                                                                                                                                                                                                                                                                                                                                                                                                                                                                                                                                                                                                                                                                                                                                                                                                                                                                                                                                                                                                                                                                                                                                                                                                                                                                                                                                                                                                                                                                                                                                                                                                                                                                                                                                                                                                                                                                                                                                                                                                                                                                                                                                                                                                                                                                                                                                                                                                                                                                                                                                                                                                                                                                                                                                                                                                                                                                                                                                                                                                                                                                                                                                                                                                                                                                                                                                                                                                                                                                                                                                                                                                                                                                                                                                                                                                                                                                                                                                                                                                                                                                                                                                                                                                                                                                                                    |
|            | observations can fully support the conclusion that DOM leached from the surface                                                                                                                                                                                                                                                                                                                                                                                                                                                                                                                                                                                                                                                                                                                                                                                                                                                                                                                                                                                                                                                                                                                                                                                                                                                                                                                                                                                                                                                                                                                                                                                                                                                                                                                                                                                                                                                                                                                                                                                                                                                                                                                                                                                                                                                                                                                                                                                                                                                                                                                                                                                                                                                                                                                                                                                                                                                                                                                                                                                                                                                                                                                                                                                                                                                                                                                                                                                                                                                                                                                                                                                                                                                                                                                                                                                                                                                                                                                                                                                                                                                                                                                                                                                                                                                                                                                                                                                                                                                                                                                                                                                                                                                                                                                                                                                                                                                                                                                                                                                                                                                         |
|            | soil layers was a dominant source                                                                                                                                                                                                                                                                                                                                                                                                                                                                                                                                                                                                                                                                                                                                                                                                                                                                                                                                                                                                                                                                                                                                                                                                                                                                                                                                                                                                                                                                                                                                                                                                                                                                                                                                                                                                                                                                                                                                                                                                                                                                                                                                                                                                                                                                                                                                                                                                                                                                                                                                                                                                                                                                                                                                                                                                                                                                                                                                                                                                                                                                                                                                                                                                                                                                                                                                                                                                                                                                                                                                                                                                                                                                                                                                                                                                                                                                                                                                                                                                                                                                                                                                                                                                                                                                                                                                                                                                                                                                                                                                                                                                                                                                                                                                                                                                                                                                                                                                                                                                                                                                                                       |
| Author     | We agree that microbial decomposition of either root derived C or also litter                                                                                                                                                                                                                                                                                                                                                                                                                                                                                                                                                                                                                                                                                                                                                                                                                                                                                                                                                                                                                                                                                                                                                                                                                                                                                                                                                                                                                                                                                                                                                                                                                                                                                                                                                                                                                                                                                                                                                                                                                                                                                                                                                                                                                                                                                                                                                                                                                                                                                                                                                                                                                                                                                                                                                                                                                                                                                                                                                                                                                                                                                                                                                                                                                                                                                                                                                                                                                                                                                                                                                                                                                                                                                                                                                                                                                                                                                                                                                                                                                                                                                                                                                                                                                                                                                                                                                                                                                                                                                                                                                                                                                                                                                                                                                                                                                                                                                                                                                                                                                                                           |
| Aution     | We agree that interodulat decomposition of either root-derived C of also inter-                                                                                                                                                                                                                                                                                                                                                                                                                                                                                                                                                                                                                                                                                                                                                                                                                                                                                                                                                                                                                                                                                                                                                                                                                                                                                                                                                                                                                                                                                                                                                                                                                                                                                                                                                                                                                                                                                                                                                                                                                                                                                                                                                                                                                                                                                                                                                                                                                                                                                                                                                                                                                                                                                                                                                                                                                                                                                                                                                                                                                                                                                                                                                                                                                                                                                                                                                                                                                                                                                                                                                                                                                                                                                                                                                                                                                                                                                                                                                                                                                                                                                                                                                                                                                                                                                                                                                                                                                                                                                                                                                                                                                                                                                                                                                                                                                                                                                                                                                                                                                                                         |
| response   | derived C may increase C values. Together with preferential sorption of C-                                                                                                                                                                                                                                                                                                                                                                                                                                                                                                                                                                                                                                                                                                                                                                                                                                                                                                                                                                                                                                                                                                                                                                                                                                                                                                                                                                                                                                                                                                                                                                                                                                                                                                                                                                                                                                                                                                                                                                                                                                                                                                                                                                                                                                                                                                                                                                                                                                                                                                                                                                                                                                                                                                                                                                                                                                                                                                                                                                                                                                                                                                                                                                                                                                                                                                                                                                                                                                                                                                                                                                                                                                                                                                                                                                                                                                                                                                                                                                                                                                                                                                                                                                                                                                                                                                                                                                                                                                                                                                                                                                                                                                                                                                                                                                                                                                                                                                                                                                                                                                                              |
|            | depleted substances, both processes account for the C pattern with soil depth, as                                                                                                                                                                                                                                                                                                                                                                                                                                                                                                                                                                                                                                                                                                                                                                                                                                                                                                                                                                                                                                                                                                                                                                                                                                                                                                                                                                                                                                                                                                                                                                                                                                                                                                                                                                                                                                                                                                                                                                                                                                                                                                                                                                                                                                                                                                                                                                                                                                                                                                                                                                                                                                                                                                                                                                                                                                                                                                                                                                                                                                                                                                                                                                                                                                                                                                                                                                                                                                                                                                                                                                                                                                                                                                                                                                                                                                                                                                                                                                                                                                                                                                                                                                                                                                                                                                                                                                                                                                                                                                                                                                                                                                                                                                                                                                                                                                                                                                                                                                                                                                                       |
|            | we discuss in line 304 to 306.                                                                                                                                                                                                                                                                                                                                                                                                                                                                                                                                                                                                                                                                                                                                                                                                                                                                                                                                                                                                                                                                                                                                                                                                                                                                                                                                                                                                                                                                                                                                                                                                                                                                                                                                                                                                                                                                                                                                                                                                                                                                                                                                                                                                                                                                                                                                                                                                                                                                                                                                                                                                                                                                                                                                                                                                                                                                                                                                                                                                                                                                                                                                                                                                                                                                                                                                                                                                                                                                                                                                                                                                                                                                                                                                                                                                                                                                                                                                                                                                                                                                                                                                                                                                                                                                                                                                                                                                                                                                                                                                                                                                                                                                                                                                                                                                                                                                                                                                                                                                                                                                                                          |
|            | Since we do not link our observations and conclusions to recent litter-derived C                                                                                                                                                                                                                                                                                                                                                                                                                                                                                                                                                                                                                                                                                                                                                                                                                                                                                                                                                                                                                                                                                                                                                                                                                                                                                                                                                                                                                                                                                                                                                                                                                                                                                                                                                                                                                                                                                                                                                                                                                                                                                                                                                                                                                                                                                                                                                                                                                                                                                                                                                                                                                                                                                                                                                                                                                                                                                                                                                                                                                                                                                                                                                                                                                                                                                                                                                                                                                                                                                                                                                                                                                                                                                                                                                                                                                                                                                                                                                                                                                                                                                                                                                                                                                                                                                                                                                                                                                                                                                                                                                                                                                                                                                                                                                                                                                                                                                                                                                                                                                                                        |
|            | alone in lines 290 to 314, we think that the response given in Referee's comment                                                                                                                                                                                                                                                                                                                                                                                                                                                                                                                                                                                                                                                                                                                                                                                                                                                                                                                                                                                                                                                                                                                                                                                                                                                                                                                                                                                                                                                                                                                                                                                                                                                                                                                                                                                                                                                                                                                                                                                                                                                                                                                                                                                                                                                                                                                                                                                                                                                                                                                                                                                                                                                                                                                                                                                                                                                                                                                                                                                                                                                                                                                                                                                                                                                                                                                                                                                                                                                                                                                                                                                                                                                                                                                                                                                                                                                                                                                                                                                                                                                                                                                                                                                                                                                                                                                                                                                                                                                                                                                                                                                                                                                                                                                                                                                                                                                                                                                                                                                                                                                        |
|            | #4 and the respective modification in the text are sufficient to clarify that we relate                                                                                                                                                                                                                                                                                                                                                                                                                                                                                                                                                                                                                                                                                                                                                                                                                                                                                                                                                                                                                                                                                                                                                                                                                                                                                                                                                                                                                                                                                                                                                                                                                                                                                                                                                                                                                                                                                                                                                                                                                                                                                                                                                                                                                                                                                                                                                                                                                                                                                                                                                                                                                                                                                                                                                                                                                                                                                                                                                                                                                                                                                                                                                                                                                                                                                                                                                                                                                                                                                                                                                                                                                                                                                                                                                                                                                                                                                                                                                                                                                                                                                                                                                                                                                                                                                                                                                                                                                                                                                                                                                                                                                                                                                                                                                                                                                                                                                                                                                                                                                                                 |
|            | this observation to DOM of different origin. We want to add here that the natural                                                                                                                                                                                                                                                                                                                                                                                                                                                                                                                                                                                                                                                                                                                                                                                                                                                                                                                                                                                                                                                                                                                                                                                                                                                                                                                                                                                                                                                                                                                                                                                                                                                                                                                                                                                                                                                                                                                                                                                                                                                                                                                                                                                                                                                                                                                                                                                                                                                                                                                                                                                                                                                                                                                                                                                                                                                                                                                                                                                                                                                                                                                                                                                                                                                                                                                                                                                                                                                                                                                                                                                                                                                                                                                                                                                                                                                                                                                                                                                                                                                                                                                                                                                                                                                                                                                                                                                                                                                                                                                                                                                                                                                                                                                                                                                                                                                                                                                                                                                                                                                       |
|            | 13 C pattern with soil depth was taken into account and used to determine                                                                                                                                                                                                                                                                                                                                                                                                                                                                                                                                                                                                                                                                                                                                                                                                                                                                                                                                                                                                                                                                                                                                                                                                                                                                                                                                                                                                                                                                                                                                                                                                                                                                                                                                                                                                                                                                                                                                                                                                                                                                                                                                                                                                                                                                                                                                                                                                                                                                                                                                                                                                                                                                                                                                                                                                                                                                                                                                                                                                                                                                                                                                                                                                                                                                                                                                                                                                                                                                                                                                                                                                                                                                                                                                                                                                                                                                                                                                                                                                                                                                                                                                                                                                                                                                                                                                                                                                                                                                                                                                                                                                                                                                                                                                                                                                                                                                                                                                                                                                                                                    |
|            | significant enrichments of the labeled samples. This was expressed in eq. 7 (lines                                                                                                                                                                                                                                                                                                                                                                                                                                                                                                                                                                                                                                                                                                                                                                                                                                                                                                                                                                                                                                                                                                                                                                                                                                                                                                                                                                                                                                                                                                                                                                                                                                                                                                                                                                                                                                                                                                                                                                                                                                                                                                                                                                                                                                                                                                                                                                                                                                                                                                                                                                                                                                                                                                                                                                                                                                                                                                                                                                                                                                                                                                                                                                                                                                                                                                                                                                                                                                                                                                                                                                                                                                                                                                                                                                                                                                                                                                                                                                                                                                                                                                                                                                                                                                                                                                                                                                                                                                                                                                                                                                                                                                                                                                                                                                                                                                                                                                                                                                                                                                                      |
|            | 204 to 210).                                                                                                                                                                                                                                                                                                                                                                                                                                                                                                                                                                                                                                                                                                                                                                                                                                                                                                                                                                                                                                                                                                                                                                                                                                                                                                                                                                                                                                                                                                                                                                                                                                                                                                                                                                                                                                                                                                                                                                                                                                                                                                                                                                                                                                                                                                                                                                                                                                                                                                                                                                                                                                                                                                                                                                                                                                                                                                                                                                                                                                                                                                                                                                                                                                                                                                                                                                                                                                                                                                                                                                                                                                                                                                                                                                                                                                                                                                                                                                                                                                                                                                                                                                                                                                                                                                                                                                                                                                                                                                                                                                                                                                                                                                                                                                                                                                                                                                                                                                                                                                                                                                                            |
| 7. Comment | Lines 360 to 364:                                                                                                                                                                                                                                                                                                                                                                                                                                                                                                                                                                                                                                                                                                                                                                                                                                                                                                                                                                                                                                                                                                                                                                                                                                                                                                                                                                                                                                                                                                                                                                                                                                                                                                                                                                                                                                                                                                                                                                                                                                                                                                                                                                                                                                                                                                                                                                                                                                                                                                                                                                                                                                                                                                                                                                                                                                                                                                                                                                                                                                                                                                                                                                                                                                                                                                                                                                                                                                                                                                                                                                                                                                                                                                                                                                                                                                                                                                                                                                                                                                                                                                                                                                                                                                                                                                                                                                                                                                                                                                                                                                                                                                                                                                                                                                                                                                                                                                                                                                                                                                                                                                                       |
|            | Could you explain where the majority of litter-derived C goes; emitted as CO 2 ?                                                                                                                                                                                                                                                                                                                                                                                                                                                                                                                                                                                                                                                                                                                                                                                                                                                                                                                                                                                                                                                                                                                                                                                                                                                                                                                                                                                                                                                                                                                                                                                                                                                                                                                                                                                                                                                                                                                                                                                                                                                                                                                                                                                                                                                                                                                                                                                                                                                                                                                                                                                                                                                                                                                                                                                                                                                                                                                                                                                                                                                                                                                                                                                                                                                                                                                                                                                                                                                                                                                                                                                                                                                                                                                                                                                                                                                                                                                                                                                                                                                                                                                                                                                                                                                                                                                                                                                                                                                                                                                                                                                                                                                                                                                                                                                                                                                                                                                                                                                                                                             |
| Author     | This is a good and very important comment/question, which we will definitely                                                                                                                                                                                                                                                                                                                                                                                                                                                                                                                                                                                                                                                                                                                                                                                                                                                                                                                                                                                                                                                                                                                                                                                                                                                                                                                                                                                                                                                                                                                                                                                                                                                                                                                                                                                                                                                                                                                                                                                                                                                                                                                                                                                                                                                                                                                                                                                                                                                                                                                                                                                                                                                                                                                                                                                                                                                                                                                                                                                                                                                                                                                                                                                                                                                                                                                                                                                                                                                                                                                                                                                                                                                                                                                                                                                                                                                                                                                                                                                                                                                                                                                                                                                                                                                                                                                                                                                                                                                                                                                                                                                                                                                                                                                                                                                                                                                                                                                                                                                                                                                            |
| response   | address. We made a detailed mass balance regarding the fate of recent litter layer-                                                                                                                                                                                                                                                                                                                                                                                                                                                                                                                                                                                                                                                                                                                                                                                                                                                                                                                                                                                                                                                                                                                                                                                                                                                                                                                                                                                                                                                                                                                                                                                                                                                                                                                                                                                                                                                                                                                                                                                                                                                                                                                                                                                                                                                                                                                                                                                                                                                                                                                                                                                                                                                                                                                                                                                                                                                                                                                                                                                                                                                                                                                                                                                                                                                                                                                                                                                                                                                                                                                                                                                                                                                                                                                                                                                                                                                                                                                                                                                                                                                                                                                                                                                                                                                                                                                                                                                                                                                                                                                                                                                                                                                                                                                                                                                                                                                                                                                                                                                                                                                     |
| F F        | C, including DOC monitoring and surface $CO_2$ monitoring. Both will be subject of                                                                                                                                                                                                                                                                                                                                                                                                                                                                                                                                                                                                                                                                                                                                                                                                                                                                                                                                                                                                                                                                                                                                                                                                                                                                                                                                                                                                                                                                                                                                                                                                                                                                                                                                                                                                                                                                                                                                                                                                                                                                                                                                                                                                                                                                                                                                                                                                                                                                                                                                                                                                                                                                                                                                                                                                                                                                                                                                                                                                                                                                                                                                                                                                                                                                                                                                                                                                                                                                                                                                                                                                                                                                                                                                                                                                                                                                                                                                                                                                                                                                                                                                                                                                                                                                                                                                                                                                                                                                                                                                                                                                                                                                                                                                                                                                                                                                                                                                                                                                                                                      |
|            | another publication (currently in preparation) which will focus on the budget in                                                                                                                                                                                                                                                                                                                                                                                                                                                                                                                                                                                                                                                                                                                                                                                                                                                                                                                                                                                                                                                                                                                                                                                                                                                                                                                                                                                                                                                                                                                                                                                                                                                                                                                                                                                                                                                                                                                                                                                                                                                                                                                                                                                                                                                                                                                                                                                                                                                                                                                                                                                                                                                                                                                                                                                                                                                                                                                                                                                                                                                                                                                                                                                                                                                                                                                                                                                                                                                                                                                                                                                                                                                                                                                                                                                                                                                                                                                                                                                                                                                                                                                                                                                                                                                                                                                                                                                                                                                                                                                                                                                                                                                                                                                                                                                                                                                                                                                                                                                                                                                        |
|            | contrast to the present publication where focus in on the fate of litter-derived OM                                                                                                                                                                                                                                                                                                                                                                                                                                                                                                                                                                                                                                                                                                                                                                                                                                                                                                                                                                                                                                                                                                                                                                                                                                                                                                                                                                                                                                                                                                                                                                                                                                                                                                                                                                                                                                                                                                                                                                                                                                                                                                                                                                                                                                                                                                                                                                                                                                                                                                                                                                                                                                                                                                                                                                                                                                                                                                                                                                                                                                                                                                                                                                                                                                                                                                                                                                                                                                                                                                                                                                                                                                                                                                                                                                                                                                                                                                                                                                                                                                                                                                                                                                                                                                                                                                                                                                                                                                                                                                                                                                                                                                                                                                                                                                                                                                                                                                                                                                                                                                                     |
|            | in soil                                                                                                                                                                                                                                                                                                                                                                                                                                                                                                                                                                                                                                                                                                                                                                                                                                                                                                                                                                                                                                                                                                                                                                                                                                                                                                                                                                                                                                                                                                                                                                                                                                                                                                                                                                                                                                                                                                                                                                                                                                                                                                                                                                                                                                                                                                                                                                                                                                                                                                                                                                                                                                                                                                                                                                                                                                                                                                                                                                                                                                                                                                                                                                                                                                                                                                                                                                                                                                                                                                                                                                                                                                                                                                                                                                                                                                                                                                                                                                                                                                                                                                                                                                                                                                                                                                                                                                                                                                                                                                                                                                                                                                                                                                                                                                                                                                                                                                                                                                                                                                                                                                                                 |
|            | To answer the question: The majority of the labeled litter. C on the one hand                                                                                                                                                                                                                                                                                                                                                                                                                                                                                                                                                                                                                                                                                                                                                                                                                                                                                                                                                                                                                                                                                                                                                                                                                                                                                                                                                                                                                                                                                                                                                                                                                                                                                                                                                                                                                                                                                                                                                                                                                                                                                                                                                                                                                                                                                                                                                                                                                                                                                                                                                                                                                                                                                                                                                                                                                                                                                                                                                                                                                                                                                                                                                                                                                                                                                                                                                                                                                                                                                                                                                                                                                                                                                                                                                                                                                                                                                                                                                                                                                                                                                                                                                                                                                                                                                                                                                                                                                                                                                                                                                                                                                                                                                                                                                                                                                                                                                                                                                                                                                                                           |
|            | indeed emitted as $CO_{(1)}$ (26.40 %) and on the other hand remained in the litter                                                                                                                                                                                                                                                                                                                                                                                                                                                                                                                                                                                                                                                                                                                                                                                                                                                                                                                                                                                                                                                                                                                                                                                                                                                                                                                                                                                                                                                                                                                                                                                                                                                                                                                                                                                                                                                                                                                                                                                                                                                                                                                                                                                                                                                                                                                                                                                                                                                                                                                                                                                                                                                                                                                                                                                                                                                                                                                                                                                                                                                                                                                                                                                                                                                                                                                                                                                                                                                                                                                                                                                                                                                                                                                                                                                                                                                                                                                                                                                                                                                                                                                                                                                                                                                                                                                                                                                                                                                                                                                                                                                                                                                                                                                                                                                                                                                                                                                                                                                                                                                     |
|            | indeed enfitted as $CO_2$ (~ 50-40 %) and on the other hand remained in the fitter                                                                                                                                                                                                                                                                                                                                                                                                                                                                                                                                                                                                                                                                                                                                                                                                                                                                                                                                                                                                                                                                                                                                                                                                                                                                                                                                                                                                                                                                                                                                                                                                                                                                                                                                                                                                                                                                                                                                                                                                                                                                                                                                                                                                                                                                                                                                                                                                                                                                                                                                                                                                                                                                                                                                                                                                                                                                                                                                                                                                                                                                                                                                                                                                                                                                                                                                                                                                                                                                                                                                                                                                                                                                                                                                                                                                                                                                                                                                                                                                                                                                                                                                                                                                                                                                                                                                                                                                                                                                                                                                                                                                                                                                                                                                                                                                                                                                                                                                                                                                                                                      |
| 0.0        | layer (~ 33-40 %).                                                                                                                                                                                                                                                                                                                                                                                                                                                                                                                                                                                                                                                                                                                                                                                                                                                                                                                                                                                                                                                                                                                                                                                                                                                                                                                                                                                                                                                                                                                                                                                                                                                                                                                                                                                                                                                                                                                                                                                                                                                                                                                                                                                                                                                                                                                                                                                                                                                                                                                                                                                                                                                                                                                                                                                                                                                                                                                                                                                                                                                                                                                                                                                                                                                                                                                                                                                                                                                                                                                                                                                                                                                                                                                                                                                                                                                                                                                                                                                                                                                                                                                                                                                                                                                                                                                                                                                                                                                                                                                                                                                                                                                                                                                                                                                                                                                                                                                                                                                                                                                                                                                      |
| 8. Comment | If so, why the older mobilizable OC did not emit as CO2?                                                                                                                                                                                                                                                                                                                                                                                                                                                                                                                                                                                                                                                                                                                                                                                                                                                                                                                                                                                                                                                                                                                                                                                                                                                                                                                                                                                                                                                                                                                                                                                                                                                                                                                                                                                                                                                                                                                                                                                                                                                                                                                                                                                                                                                                                                                                                                                                                                                                                                                                                                                                                                                                                                                                                                                                                                                                                                                                                                                                                                                                                                                                                                                                                                                                                                                                                                                                                                                                                                                                                                                                                                                                                                                                                                                                                                                                                                                                                                                                                                                                                                                                                                                                                                                                                                                                                                                                                                                                                                                                                                                                                                                                                                                                                                                                                                                                                                                                                                                                                                                                                |
| Author     | Older OC definitely did emit as $CO_2$ (katabolic pathway), or is recycled by soil                                                                                                                                                                                                                                                                                                                                                                                                                                                                                                                                                                                                                                                                                                                                                                                                                                                                                                                                                                                                                                                                                                                                                                                                                                                                                                                                                                                                                                                                                                                                                                                                                                                                                                                                                                                                                                                                                                                                                                                                                                                                                                                                                                                                                                                                                                                                                                                                                                                                                                                                                                                                                                                                                                                                                                                                                                                                                                                                                                                                                                                                                                                                                                                                                                                                                                                                                                                                                                                                                                                                                                                                                                                                                                                                                                                                                                                                                                                                                                                                                                                                                                                                                                                                                                                                                                                                                                                                                                                                                                                                                                                                                                                                                                                                                                                                                                                                                                                                                                                                                                                      |
| response   | microorganisms and consequently used as a source to build-up biomass (anabolic                                                                                                                                                                                                                                                                                                                                                                                                                                                                                                                                                                                                                                                                                                                                                                                                                                                                                                                                                                                                                                                                                                                                                                                                                                                                                                                                                                                                                                                                                                                                                                                                                                                                                                                                                                                                                                                                                                                                                                                                                                                                                                                                                                                                                                                                                                                                                                                                                                                                                                                                                                                                                                                                                                                                                                                                                                                                                                                                                                                                                                                                                                                                                                                                                                                                                                                                                                                                                                                                                                                                                                                                                                                                                                                                                                                                                                                                                                                                                                                                                                                                                                                                                                                                                                                                                                                                                                                                                                                                                                                                                                                                                                                                                                                                                                                                                                                                                                                                                                                                                                                          |
|            | pathway). Microbial decomposition is also the primary reason for the strong